# Applications of Spatial Transcriptomics in Veterinary Medicine: A Scoping Review of Research, Diagnostics, and Treatment Strategies

**DOI:** 10.3390/ijms26136163

**Published:** 2025-06-26

**Authors:** Rachael M. Weiderman, Mahamudul Hasan, Laura C. Miller

**Affiliations:** Diagnostic Medicine and Pathobiology, College of Veterinary Medicine, Kansas State University, 1800 Denison Ave, Manhattan, KS 66506, USA; warhurst@ksu.edu (R.M.W.); mahamudul@vet.k-state.edu (M.H.)

**Keywords:** spatial, transcriptomics, veterinary medicine

## Abstract

Spatial transcriptomics is an emerging technology that maps gene expression within tissue architecture. Its expanding use in medicine and veterinary science supports research, precision diagnostics, biomarker discovery, and development of targeted treatment strategies. While spatial transcriptomics applications in human health are well-documented with significant publication diversity and volume, published applications in veterinary medicine remain limited. A comprehensive search of PubMed was conducted, focusing on studies published from 2016 to early 2025 that employed spatial transcriptomics in the context of disease research, diagnosis, or treatment in human or animal health. The review followed the Arksey and O’Malley framework and adhered to Preferred Reporting Items for Systematic Reviews and Meta-Analyses Extension for Scoping Reviews (PRISMA-ScR) guidelines. A total of 1398 studies met the inclusion criteria. The studies highlighted emerging trends of comparative research with animal model use for human health research. Commonly used spatial transcriptomics platforms included 10× Visium, Slide-seq, Nanostring (GeoMx, CosMX), and multiplexed error-robust fluorescence in situ hybridization (MERFISH). Key gaps in publications include limited veterinary representation, interspecies comparisons, standardized methods, public data use, and therapeutic studies, alongside biases in disease, species, organ, and geography. This review presents the current landscape of spatial transcriptomics publications for human and animal research and medicine, providing comprehensive data and highlighting underrepresented research areas and gaps for future consideration.

## 1. Introduction

Spatial transcriptomics (ST) is an advanced molecular profiling technology that enables the quantification and mapping of gene expression in intact tissue sections across biological systems. Unlike bulk RNA sequencing or single-cell RNA sequencing (scRNA-seq), which provide high-throughput transcriptional data without spatial context, ST preserves the architecture of tissues while providing insights into gene expression [1,2]. This allows researchers to study gene expression patterns, tissue heterogeneity, host-immune response, and cellular interactions across sample microenvironments, offering novel insights into tissue organization, disease mechanisms, and cell signaling networks [3,4,5,6]. Since introduction into the literature in 2016, ST has emerged as a transformative tool in both biomedical research and clinical applications [1]. In human medicine, ST has been widely applied to research studying tumor microenvironments [5,7,8], identifying biomarkers for diagnosis and therapeutics [8,9,10], and uncovering molecular mechanisms of disease [1,2,3,4,5,6,9,10,11,12,13,14]. Innovations including Slide-seq [15], multiplexed error-robust fluorescence in situ hybridization (MERFISH) [16], and high-throughput commercial platforms including 10× Visium [17] and GeoMx Digital Spatial Profiler [18] have further accelerated adoption of ST across research and clinical specialties by increasing accessibility and removing technical barriers.

Although most literature within the field answers questions about human health and disease, ST has gained traction in veterinary medicine, with emerging studies exploring applications for companion animal health [19], livestock medicine [20], and zoonotic disease research (COVID-19) [21]. ST applications in veterinary medicine have insights into mucosal immunity related to veterinary medicine and animal health [22], nervous system degeneration [19], and parasitology [23]. Although limited in volume, recent veterinary medicine publications support enhanced breadth and depth of research in this field, providing novel data for improved animal health and new opportunities for cross-species comparisons [21]. ST is particularly well-suited for advancing such work, as it enables spatially resolved mapping of gene expression within intact tissues. This allows for precise alignment of molecular signatures across anatomically homologous regions in different species, facilitating the identification of both evolutionarily conserved pathways and species-specific molecular features. Such comparative insights are essential for bridging translational research between animal models and human disease [24].

However, despite rapid technological and analytical advances, and emerging applications in veterinary medicine, the overall publication landscape of ST research across diverse species and rare disease models remains fragmented. As such, a comprehensive mapping of current applications, technological trends, and methodological gaps is both timely and critical. To our knowledge, no comprehensive synthesis has yet been conducted that integrates ST applications across both human and veterinary contexts or evaluates specific trends in diagnostic, therapeutic, and mechanistic research.

This scoping review aims to systematically assess how ST is currently applied in human and veterinary medicine, with emphasis on diagnostic, therapeutic, and mechanistic contributions. This review bridges existing gaps in veterinary ST research and proposes a roadmap for advancing cross-species translation.

### 1.1. Rationale

Despite growing interest, rapid advancement, and significant volume increase in publications across the field, the literature remains disproportionately focused on human samples and specific disease models, including oncology and neurobiology [9,25,26,27,28,29]. This fragmentation has led to underrepresentation of species, veterinary and zoonotic diseases, and comparative medicine knowledge, limiting cross-species and One Health benefits. Lack of comprehensive syntheses examining the broader use of ST across biological systems, particularly for diagnostic and therapeutic potential in veterinary medicine, zoonotic research, and comparative pathology, represents a significant gap in the current literature. This scoping review aims to highlight the valuable insights gained from early ST field publications in human and veterinary health and provide discussion and limitations in the field for future research opportunities.

To address this gap, it is important to critically examine how ST has been applied across both human and animal health contexts. While animal models are used frequently in ST applications to study disease, we have not fully realized potential in comparative medicine, comparative pathology, zoonotic disease research, and disease surveillance. This presents a critical gap, especially given the unique potential of ST to bridge human and animal health research [30,31].

Given the novelty and rapid evolution within the field, this scoping review offers a timely, appropriate methodological approach to examine human and veterinary health applications. Unlike systematic reviews conducted to answer defined clinical questions, scoping reviews map the full breadth and depth of a field, clarifying emerging concepts, providing landmark accomplishments, and identifying gaps in knowledge and application. Therefore, conducting a scoping review for ST applications in human and veterinary medicine provides the greatest opportunity to fully capture the current state of research. To align with best practices, this review follows the Arksey and O’Malley framework and adheres to PRISMA-ScR (Preferred Reporting Items for Systematic Reviews and Meta-Analyses extension for Scoping Reviews) guidelines. This ensures the transparency, rigor, and reproducibility of our findings.

### 1.2. Objectives

This scoping review aims to systematically map the current landscape of ST applications in human and veterinary health, with an emphasis on diagnostic, therapeutic, and mechanistic insights. Specific objectives are to

Identify and characterize published studies that utilize ST to investigate disease processes in human and veterinary contexts;Summarize thematic trends across the literature, including types of diseases studied, species represented, spatial platforms and analysis tools employed, and integration with complementary technologies such as single-cell RNA sequencing and imaging;Compare applications of ST in human and veterinary research, highlighting patterns, applications, and underexplored areas;Identify knowledge gaps, technical challenges, and future opportunities to expand ST within translational and comparative biomedical research.

These objectives align with the rationale of the review by providing a comprehensive snapshot of the ST field.

### 1.3. Conceptual Framework: Population, Concept, Context (PCC)

To guide the scope, eligibility criteria, and data charting strategy of this review, we applied the PCC framework—a structured model recommended for scoping reviews. The PCC framework defines the population, concept, and context, which helped us systematically determine study inclusion and organize extracted data.

Population: We included studies involving human or animal subjects—such as companion animals, livestock, and experimental disease models—investigated in the context of health or disease. This criterion shaped the inclusion of species-specific research and excluded purely in vitro or plant-based studies;Concept: Eligible studies were required to apply ST as a central methodological approach for profiling gene expression in situ. Studies using bulk or single-cell RNA-seq without spatial resolution were excluded. This ensured conceptual focus on ST technologies and their diagnostic, mechanistic, or therapeutic applications;Context: Studies had to be conducted within a biomedical or veterinary research context, covering disease-relevant themes such as cancer, infectious diseases, neurological diseases, or immune-mediated conditions. This context criterion informed both inclusion boundaries and the organization of results by disease type.

By applying the PCC framework in this way, we ensured that our inclusion criteria were tightly aligned with the study objective. The same structure also guided data charting, allowing us to categorize included articles based on subject population, spatial method used, and research context.

## 2. Methods

### 2.1. Framework

This scoping review was based on the Arksey and O’Malley framework [32], refined by Levac et al. (2010) and reported using PRISMA-ScR guidelines [33]. These frameworks and guidelines represent the best practices for scoping reviews. The PRISMA-ScR flow diagram providing a summary of the literature review is visualized in Figure 1.

### 2.2. Eligibility Criteria

All studies were reviewed for predefined eligibility criteria to determine inclusion or exclusion status from the scoping review. Eligibility criteria were designed to capture high-quality, relevant publications aligned with review objectives.

Studies were included if they met the following criteria: (1) the research was conducted in the context of animal or human health, focusing on disease diagnosis, pathology, or treatment; (2) the publication was peer-reviewed; (3) the article was an original research study, systematic review, or meta-analysis; (4) the study was published between 2016 and 7 February 2025; (5) the study involved animal models, human tissue samples, or publicly available datasets from human or veterinary sources; (6) ST was used as a central methodology and not merely mentioned; and (7) the study reported biological, clinical, or methodological insights relevant to diagnosis, pathology, or treatment.

For veterinary-specific studies, “disease relevance” was operationalized to include naturally occurring disease conditions in companion animals and livestock, zoonotic diseases involving veterinary species, and production-limiting health issues relevant to animal agriculture. Additionally, studies involving experimentally induced diseases in animal models were included when ST was used to generate clinical, biological, or translational insight. This operational definition was developed in accordance with scoping review methodology, which allows tailoring of inclusion criteria to the conceptual focus of the review [34].

Studies were excluded if they met any of the following criteria: (1) they did not utilize ST methodologies—such as studies relying solely on bulk or single-cell RNA sequencing without spatial resolution; (2) they were outside the scope of veterinary or human health and disease, including studies focused solely on method development without new biological or clinical insights; (3) they were not peer-reviewed, including grey literature sources such as preprints, conference abstracts, and theses, which were excluded to maintain the scientific rigor of the review; (4) they were case reports or scoping reviews, which were excluded due to the lack of standardized methodology and comparative analytical depth required for inclusion in this synthesis; (5) they were not published in English, as language restrictions were applied to ensure accurate interpretation and consistent evaluation of study content. All inclusion and exclusion criteria are visualized in Table 1. These criteria were designed to capture a broad yet focused set of studies applying ST in clinically and translationally relevant settings across species.

### 2.3. Study Selection and Screening Summary

The full search strategy, database sources, screening process, and data charting methodology are provided in Appendix A. These steps followed established scoping review procedures to ensure methodological rigor and reproducibility.

## 3. Results

This section presents the findings from the scoping review, beginning with an overview of the included studies and followed by a synthesis of descriptive and thematic trends. We summarize the scope, origin, and methodological characteristics of the 1398 included studies, including their disease focus, species studied, and types of ST platforms and analytical tools used. The results are organized to highlight how ST have evolved over time and how different species, model systems, and data sources are represented. These findings provide a comprehensive view of the current landscape of ST across human and veterinary biomedical research.

### 3.1. Overview of Included Studies

#### 3.1.1. Selection of Sources of Evidence

A total of 3125 records were identified through PubMed. After removing 100 duplicates, 3025 titles and abstracts were screened in Rayyan AI. Of these, 582 were excluded for not meeting inclusion criteria during abstract review. The remaining 2443 full texts were sought for retrieval, and 2331 were reviewed, with 933 excluded for reasons such as absence of ST methods or lack of disease relevance. In total, 1398 studies were included in the final synthesis. The selection process is outlined in the PRISMA-ScR Flow Diagram (Figure 1).

#### 3.1.2. Characteristics of Included Studies

In total, 1398 included studies spanned a publication period from 2016 to early 2025, reflecting the growing adoption of ST since its formalization [1]. Most ST publications focused on human tissues, especially in oncology, with common targets including the brain, breast, lung, liver, and gastrointestinal tract. A smaller but notable subset investigated veterinary or comparative models, such as mice, macaques, cattle, and canines, often in the context of infectious diseases, zoonotic diseases, neurodegeneration, or translational cancer research. However, most animal research in this subset employed animal models for human health insights.

Studies employed a wide range of ST platforms—most commonly 10× Genomics Visium, followed by Slide-seq, MERFISH, and sequential fluorescence in situ hybridization (seqFISH). Analytical tools varied, with Seurat (v5.0), Giotto (v1.3), BayesSpace, and SpaGCN most frequently used for spatial data integration, clustering, and domain detection. Some tools represented emerging or specialized approaches, reflecting the evolving nature of analytical frameworks in the ST field.

Cross-platform data integration presented substantial challenges, particularly when combining datasets from platforms with varying spatial resolution, tissue compatibility, and transcript capture efficiencies. For example, Visium supports moderate-resolution profiling (~55 μm spots) and accommodates both formalin-fixed/paraffin-embedded (FFPE) and fresh-frozen tissues, while Slide-seq and MERFISH offer near-single-cell resolution (~10 μm or better) but are typically limited to fresh-frozen samples [35,36]. These differences introduced batch effects and inconsistencies in transcript quantification, spatial architecture, and alignment across samples or studies. To address these issues, studies commonly implemented batch correction and integration pipelines using tools such as Seurat v5, which offers integration methods via functions like IntegrateLayers that accommodate multiple batch-correction strategies, and Harmony, which aligns latent embedding spaces across samples and platforms [37]. However, these tools have limitations, particularly when integrating data from fundamentally different ST technologies (e.g., Visium vs. Slide-seq), which differ in resolution, capture chemistry, and coordinate systems. To overcome such challenges, recent computational methods have been proposed. For instance, SPIRAL introduces a graph–domain adaptation framework combined with cluster-aware optimal transport to align both gene expression and spatial coordinates across heterogeneous platforms [38]. Similarly, uniPort, initially developed for single-cell multi-omics integration, was adapted for ST to align datasets across different technologies by applying optimal transport principles to correct for batch effects [39].

Articles were also classified based on their primary application focus, including diagnostic, mechanistic, or therapeutic. Overall, the included studies provide a representative sample within the total literature in the field of ST, revealing a strong research preference toward human health applications and emerging investigations in veterinary medicine and animal health.

#### 3.1.3. Results of Included Studies

Individual studies were summarized based on shared characteristics and thematic categories. Key findings from included articles were synthesized and grouped by disease focus, ST platform, and analytical approach. Human oncology studies were the most frequent, with subtypes including breast, liver, colorectal, brain, and lung cancers investigated using Visium, Slide-seq, and multi-modal methods.

Studies in infectious diseases and cardiac, reproductive, metabolic, and neurodevelopmental diseases were less common but increasing over time. Veterinary and zoonotic studies comprised a smaller portion of the literature and primarily focused on model species and One Health-related applications. Each thematic section of the results presents representative studies, platforms, and findings to illustrate the diversity of approaches and applications across the field. In Section 4 provide specific subsets of results in greater detail are provided for the application areas of ST.

#### 3.1.4. Synthesis of Results

Findings were synthesized using both quantitative descriptive summaries and thematic categorization. Descriptive statistics were used to quantify trends in publication year, disease focus, species studied, ST platforms, and analytical tools. These data were presented using graphs and tables to visualize frequency and distribution patterns across variables such as technology adoption, species (human versus animal model), cancer subtypes, and organ systems studied. Studies were also categorized by central application type, including diagnostic, therapeutic, or mechanistic. Thematic synthesis was applied to identify patterns and emerging trends across the volume of literature. This mixed-method synthesis approach enabled a comprehensive mapping of how ST is currently being applied across disciplines and highlighted emerging gaps in underrepresented diseases, species, and technologies.

### 3.2. Descriptive Trends

This section summarizes key descriptive trends observed across included publications. These trends reflect the evolution of the field from methodology to a broadly applied research platform. We discuss findings including annual publication growth, geographic distribution, species use, and model systems to describe global applications of ST.

#### 3.2.1. Publications by Year

The rapid growth in ST publications reflects the importance of spatial context in gene expression analysis, connecting tissue structure and function in health and disease. The field gained momentum following the landmark 2016, introducing the first high-throughput method for capturing spatially resolved transcriptomic data [1].

The commercial release of the Visium platform by 10× Genomics contributed greatly to the drastic increase in publication volume from 2019 to 2020, making ST applications more accessible by lowering technical barriers [1,3,4]. This fueled technology adoption in both academic and clinical settings. The Visium release aligned with the COVID-19 pandemic, which further accelerated ST research [21,40,41,42].

Major research consortia, including the Human Cell Atlas and BRAIN Initiative, also began adoption of ST technologies around this time to create reference atlases of healthy and diseased tissues across organs and species [43,44,45,46,47,48]. These reference maps are used in several studies; most publications generate novel spatial data.

These trends indicate a shift from specialized research methods to accessible technology for studying health and disease across biological systems and microenvironments. Figure 2 visualizes these upward trends in publication volume for studies meeting inclusion criteria, including partial data represented for 2025 through 7 February.

#### 3.2.2. Human Versus Animal Model Use

ST has been widely applied across human and animal models, demonstrating versatility for investigating disease mechanisms, cellular architecture, and tissue organization in a variety of biological systems. Among the 1398 studies included in this review, 1062 studies employed human models and 619 employed animal models (Figure 3).

Animal models play a foundational role in ST research, specifically for platform validation and mechanistic and developmental biology applications. Landmark platform validation studies using murine samples represent a well-documented area in the literature, introducing high-resolution technologies for precise cellular mapping of complex tissues [1,3,4,16,17,18]. In ST, the use of animal models often represents a method for study of human disease, while the use of animal models for ST discovery in veterinary medicine contexts remains limited to few health and disease states [19,20,21,22,23]. This bias toward human disease research represents an opportunity for increasing research and diversity in veterinary medicine and animal health topics to support comparative pathology and One Health initiatives.

Human studies are conducted to drive direct clinical outcomes in precision medicine and reveal diagnostic insights for diseases of clinical importance. For example, Maynard et al. (2021) mapped spatial gene expression across cortical layers in the human dorsolateral prefrontal cortex to provide insight into neuropsychiatric diseases [49]. Given the well-established relationship between structure and function in the human brain, Maynard’s map represents a landmark achievement toward understanding diseases originating from specific cortical layers. Additional studies toward clinical outcomes include Hirz et al.’s (2023) research in prostate cancer and Guo et al.’s (2023) research in cervical cancer, illustrating the clinical utility of spatial profiling across specific cancers toward biomarker discovery and improved diagnosis and treatment [50,51].

Translational and cross-species studies employing ST technologies are increasing in volume. Exploration of conserved genes across species provides us critical insights toward understanding the evolutionary divergence of specific cell types [52], which can be applied to understanding normal development, developmental, and disease states for various biological systems. Overall, use of human and animal models to study gene expression and tissue heterogeneity across health and disease states represents a commitment to a deeper understanding of biological mechanisms, improved diagnostic precision, and the development of targeted therapeutic strategies across diverse species and conditions. As ST platforms become increasingly accessible and multimodal, their utility across research and clinical questions will increase, with opportunities to prioritize inclusion of diverse species and models toward comparative and translational insights. Specifically, further expanding research into targeted veterinary and zoonotic disease models supports the advancement and adoption of ST as a transformative tool in global health.

#### 3.2.3. Species Studied

ST applications have been employed across a diverse range of species, underscoring the broad applicability and adaptability of technology across biological systems. Among 1398 included studies, human tissues were most frequently studied, being referenced in 1062 publications. Murine models were the predominant animal models, emphasizing their critical role in translational research for human health and disease. Additional species models employed in ST research include zebrafish, rats, and smaller numbers of non-human primates, pigs, rabbits, and other species.

Mice are a cornerstone model animal in ST research due to genetic tractability for health insights, tissue compatibility with spatial technologies, and biological similarity to humans. Murine models have been instrumental in validating high-resolution technologies including Slide-seqV2 [53] and Seq-Scope [54]. Additionally, research with murine models has supported critical insights for human health processes including cardiac remodeling, tumor progression, and neurodegeneration [55,56].

Publications studying companion animals and livestock represent an emerging and underrepresented area within the literature. Increasing investment in studying these species with ST technologies is critical toward increasing knowledge in comparative biology and veterinary medicine and reducing research biases favoring specific model species and diseases. Currently, enhanced insights from studying companion animals using ST technologies include cancer immunotherapy in dogs [57], with opportunities to expand research models to study additional companion animal species and livestock health and productivity. Non-mammalian species are also emerging as models in the literature, especially for key studies surrounding biological development. Zebrafish are a notable species offering unique advantages for studying embryogenesis [58] and cardiac system evolution [59] due to their genetic profiles and optical density. Zebrafish studies provide novel insights for many biological questions centered around development across vertebrate species.

Human studies primarily emphasize translational research focuses, bridging scientific methods and clinical applications for improved health outcomes. Landmark publications with human models include the pioneering publication studying gene expression that formally introduced ST [1], the first study to combine ST and scRNA-seq to study cancer [51], the first ST atlas of the human brain [49], and one of the first high-resolution maps of human bone [60]. Insights from biomedical ST research have improved our understanding of disease at subcellular levels, enhancing our capabilities to prevent, diagnose, and treat.

Non-human primates have also represented models of human disease in ST research, with research providing critical insights into neuropsychiatric diseases [61], brain cellular diversity [62], major depressive disorders [63], and neurodevelopmental diseases [61]. However, non-human primates are used as biological models less frequently than other animal species, such as murine models.

In summary, ST technologies have advanced our understanding of gene expression and tissue architecture across various biological systems. Human and murine models currently dominate the literature; however, the recent inclusion of diverse species supports cross-species comparisons, enhances translational insights, improves animal health, and supports One Health-oriented frameworks. Expanding species representation and interdisciplinary, collaborative study supports further leveraging of ST toward fuller potential for biomedical sciences and veterinary medicine. Figure 4 summarizes species diversity across the included literature to highlight current trends and gaps for future research consideration.

## 4. Application Areas of ST

This section explores how ST is being used across a range of research applications and organ systems. To better understand the focus of each study, we grouped publications by their primary purpose, including mechanistic research, clinical diagnosis, and therapy development. This categorization supports quantification and clarity for specific contributions of ST, highlighting distinct biological insights, disease mechanisms, and clinical applications across biomedical and veterinary medical contexts.

### 4.1. Diagnosis, Therapeutics, and Research Applications

ST is being used across a growing range of research applications, from foundational biology to clinical translation. To better understand these trends, we categorized the included studies based on their primary research purpose. These categories reflect the current technological maturity of spatial platforms and the varied ways researchers are leveraging spatial data to explore disease mechanisms and support diagnostic and therapeutic strategies. This section outlines the distribution of studies across these categories and highlights representative work from each domain.

#### Publications by Application and Central Purpose

ST studies are broadly classified into three categories: mechanistic (the largest group), which includes investigations of immunological responses, cell signaling, development, and other foundational biological processes; therapeutic, encompassing drug response profiling, treatment stratification, and personalized medicine; and diagnostic, focusing on biomarker discovery, disease subtyping, and syndrome-linked profiling. Although mechanistic studies currently dominate the field, therapeutic and diagnostic applications—particularly in oncology—are expanding rapidly, underscoring the translational potential of ST technologies.

Mechanistic studies represent the largest research investment, investigating spatial patterns of gene expression to understand biological processes including cell signaling, tissue organization, host-immune response, and organism development. Rather than aiming for direct clinical application, these studies focus on uncovering foundational biology that underlies disease and tissue function. For example, researchers have explored tumor heterogeneity and immune microenvironments in cancer and inflammatory disease [1,2,7,8,23,28,48] as well as neurodegenerative conditions [19,25,43,56,58]. Mechanistic studies provide the basis for future research and applications in therapeutic and diagnostic innovation.

Therapeutic studies use ST to identify drug targets, evaluate treatment responses, and guide personalized medicine strategies. These applications are especially common in oncology, where spatial profiling can identify therapy-resistant tumor niches or help stratify patients by molecular phenotype [28,64]. Some examples of therapeutic studies include examination of spatial remodeling in pancreatic cancer following therapy [64] and profiling of immune and stromal niches, contributing to therapy heterogeneity [28,48,50]. Other studies focus on cardiac, liver, and neurological disease models, using ST data to improve the specificity and efficacy of treatment approaches toward precision medicine [55,57,60].

Diagnostic studies apply ST technologies to research improved disease classification, discover biomarkers, and guide patient stratification. These studies combine spatial gene expression with pathology or imaging data to reveal distinct cellular neighborhoods and tissue states. Diagnostic research applications have shown promise in cancer subtyping [1,7,8,23,28,51], infectious and dermatologic diseases [21,22,40,42,65], and liver or gastrointestinal diseases [7,23,64,66]. In many cases, spatial biomarkers offer greater specificity than bulk or single-cell approaches alone, particularly when linked to a histological context. This makes spatial biomarkers a preferred diagnostic tool for ensuring specificity and sensitivity of diagnostic methods.

Across the included literature, ST has been applied to a wide range of biological and clinical questions. Mechanistic research is the most common application within the field, with studies focusing on spatial gene regulation, cell-to-cell interactions, and microenvironmental architecture. Therapeutic applications were identified in studies particularly in oncology and immune modulation. Diagnostic applications and platform validation studies are central publication types within the field. Exploratory publications generate atlases or foundational data in new tissues or disease models for research and clinical use.

Importantly, ST is increasingly influencing clinical and translational research. Research has mapped spatial patterns of cervical cancer progression [51], and immunosuppressive niches have been identified in prostate tumors [50]. These studies illustrate how spatial technologies can inform precision oncology and patient-specific decision making toward improved health outcomes across disease subtypes.

Each of these study types contributes to the growing field of ST across human and veterinary contexts. Visualizing the range and frequency of these applications, as shown in Figure 5, provides a clearer picture of how the field is evolving and where new research opportunities are emerging.

### 4.2. Organ Systems Studied

ST has been applied across a wide variety of organ systems due to the unique ability to preserve tissue architecture while capturing gene expression within spatial context. Among published studies, the digestive system emerged as the most frequently analyzed organ system (208 articles), followed by the nervous system (189), reproductive system (183), immune system (125), and integumentary system (76). The prominence of brain-focused research reflects the critical role of spatial organization in neural development, disease progression, and neuropsychiatric function, representing research questions explored.

In a landmark study, Maynard et al. (2021) applied ST to map gene expression across cortical layers of the human dorsolateral prefrontal cortex, revealing transcriptional patterns linked to neurodevelopmental and psychiatric conditions [49]. Similarly, Stickels et al. (2021) utilized Slide-seqV2 to achieve near-cellular resolution in the mouse hippocampus to enable reconstruction of fine-scale neural circuits [53]. These studies exemplify the capacity of ST to resolve complex tissue organization and intercellular communication.

In oncology research, ST has been widely adopted to study organs aligned with the most common cancer types, including breast, lung, liver, and pancreas, due to the necessity of understanding tumor heterogeneity and immune microenvironments for cancer diagnosis and treatment. High-resolution spatial mapping has been employed to define molecular subtypes in pancreatic and liver tumors, advancing therapeutic stratification strategies [5,7,64,66]. In pulmonary studies, spatial profiling has uncovered immune and epithelial remodeling in response to SARS-CoV-2 infection [21,40,42], and liver research has mapped zoned patterns of regeneration and fibrosis in murine models [7]. Spatial studies of the skin are expanding rapidly, with recent publications characterizing spatial immune architecture in gingival and epidermal tissues [51,67].

Critical research questions about organ structure, function, and disease are increasingly addressed through ST across all major body systems. Organ selection in these studies often reflects both biological significance and practical considerations such as clinical relevance, sample accessibility, and anatomical complexity. The diversity of organ systems studied in ST is illustrated in Figure 6.

### 4.3. Human Disease Applications

#### 4.3.1. Cancer

Cancer represents the most extensively studied application of ST, driven by the spatial heterogeneity of tumors and their surrounding microenvironments (Figure 7). ST maintains intact tissue architecture while resolving gene expression at spatially defined loci, enabling detailed characterization of tumor subclones, immune cell niches, and stromal organization. This technology has been widely applied to solid tumors, including breast, lung, liver, brain, colon, pancreas, and prostate [1,5,7,8,23,28].

Notable studies have identified stromal resistance zones in colorectal cancer [41,62] and region-specific microglial suppression in glioblastoma [11,25]. Platforms such as 10× Genomics Visium, Slide-seqV2, and Seq-Scope have been central to this progress, providing spatial resolution ranging from spot-level [1] to near-cellular or subcellular precision [53,54]. Integration of spatial data with single-cell RNA sequencing and machine learning has further refined tumor subtype classification, response prediction, and biomarker discovery—accelerating the advancement of precision oncology [5,7,8,23,28,29,64].

#### 4.3.2. Metabolic Disease

Metabolic diseases such as non-alcoholic fatty liver disease (NAFLD), type 2 diabetes, and obesity are characterized by regionally organized tissues including the liver, pancreas, and adipose tissue, making them ideal candidates for ST analysis. ST facilitates the investigation of tissue remodeling and metabolic reprogramming during disease progression by preserving zonal architecture and enabling high-resolution gene expression mapping.

In murine models, Liu et al. (2022) demonstrated age-associated spatial changes in hepatocyte gene expression, revealing zoned alterations in metabolic pathways and regenerative capacity [58]. Clinical and translational applications of ST are expanding, with recent studies uncovering tumor emergence within fibrotic liver zones [7], and mucosal–immune coupling in human gingival tissue, reflecting spatial disruption in barrier and immune function [67]. These findings underscore the utility of ST in detecting early metabolic imbalances and identifying therapeutic targets.

#### 4.3.3. Infectious Diseases

ST has significantly advanced infectious disease research by enabling the spatial localization of immune responses, tissue injury, and pathogen distribution within intact tissue architecture. In the context of COVID-19, Delorey et al. (2021) developed multi-organ spatial atlases capturing viral presence and immune cell dynamics across respiratory, cardiac, and gastrointestinal tissues [68].

In tuberculosis, ST has been employed to map immune cell distribution within granulomas, revealing compartmentalized inflammatory and fibrotic regions [69]. Additional studies on tuberculosis infections have characterized spatially distinct immune microenvironments and host–pathogen interfaces, offering insights into local immune evasion strategies and pathogen persistence [65,69]. Collectively, these spatial approaches are enhancing the development of immune-targeted therapies and diagnostic strategies that are informed by tissue context.

#### 4.3.4. Gastrointestinal Diseases

Gastrointestinal diseases such as inflammatory bowel disease (IBD) and colorectal cancer (CRC) exhibit regionally specific gene expression patterns that ST can resolve. In CRC, ST revealed chemotherapy-resistant fibroblast populations in tumor margins [48,70]. The growing role of ST in translational gastrointestinal research aims toward improved understanding of disease mechanisms, diagnostic tools, and therapeutic approaches.

#### 4.3.5. Cardiac Diseases

ST is increasingly being applied to cardiac diseases, including myocardial infarction (MI), cardiac fibrosis, and cardiac tumors, where it enables spatially resolved gene expression profiling across infarct zones, border regions, and remote myocardium. In MI models, Yamada et al. (2022) demonstrated spatial gradients of immune fibroblast signaling within fibrotic zones, providing mechanistic insight into post-infarction remodeling [55].

Additional datasets from a study of tumor-associated microenvironments have revealed spatially distinct angiogenic, inflammatory, and stromal gene programs [64]. As human cardiac tissue atlases expand, ST is poised to play a central role in both cardiovascular disease modeling and precision cardiology.

#### 4.3.6. Neurodevelopmental Disorder

ST is becoming a powerful tool in the study of neurodevelopmental disorder, such as autism spectrum disorder (ASD), epilepsy, and intellectual disability, which are characterized by spatial–temporal dysregulation of gene expression during critical stages of brain development. ST enables detection of early regional transcriptional changes in the brain, as illustrated in the cortical atlas developed by Maynard et al. (2021) (Section 4.2), offering insights into neurodevelopmental pathogenesis [49].

Animal models and mouse brain atlases remain essential for probing spatial dysregulation in both neuronal and glial compartments [25,44,46]. ST enables detection of early regional transcriptional changes that drive long-term neurological outcomes, thereby offering critical insights into the pathogenesis of neurodevelopmental disorder.

#### 4.3.7. Reproductive Diseases

ST is advancing reproductive health research by enabling high-resolution mapping of transcriptional programs across anatomically distinct regions of the uterus, placenta, ovary, and cervix. Diseases such as endometriosis, infertility, preeclampsia, and gynecologic cancers have been profiled using spatial technologies, revealing cell-type-specific and regionally confined gene expression patterns [6,51].

The use of FFPE-compatible platforms such as 10× Genomics Visium has further enabled the application of ST to clinical and archival reproductive tissues, facilitating its integration into translational and diagnostic research [1,5,8].

#### 4.3.8. Rare and Orphan Diseases

Rare diseases such as Rett syndrome, tuberous sclerosis, fibrolamellar carcinoma, and juvenile scleroderma often exhibit spatially complex tissue architectures that challenge conventional profiling methods. ST enables the detection of localized immune responses, transcriptional gradients, and cell–cell interactions that are critical to understanding disease pathogenesis.

In neurodevelopmental diseases such as Rett syndrome, previously discussed in Section 4.2, Maynard et al. (2021) illustrated how ST revealed laminar gene expression and region-specific microglial activation, offering insight into early disruptions in cortical development [49].

In oncology, ST has been applied to map tumor heterogeneity, treatment resistance, and immune infiltration patterns in rare pediatric cancers, including those with limited histological biomarkers [5,6,23,28]. These applications underscore the importance of ST in biomarker discovery, disease classification, and elucidation of rare disease microenvironments.

#### 4.3.9. Psychiatric Disorder

Psychiatric disorders, including schizophrenia, major depressive disorder (MDD), and bipolar disorder, involve spatially defined disruptions in brain circuitry and gene expression. ST enables high-resolution profiling of these disruptions within the structural framework of the brain. Building on prior spatial brain mapping efforts (see Maynard et al., 2021, in Section 4.2), recent work integrates ST with imaging and multi-omics to clarify the molecular basis of psychiatric diseases [49].

ST data with imaging modalities and multi-omic platforms have further deepened our understanding of how molecular perturbations manifest as behavioral symptoms, informing the development of spatially guided therapeutic interventions for psychiatric disorders [2,6,49]. Figure 8 summarizes the frequency of publications within each disease category reviewed in this section.

### 4.4. Veterinary and Zoonotic Applications

Although ST initially emerged from human biomedical research applications [1], its use has expanded significantly in recent years across veterinary and zoonotic disease models [19,20,30]. While many studies involving animal tissues utilize them as proxies for human pathophysiology, a growing number of publications now focus on livestock, companion animals, and wildlife species, highlighting ST’s capacity to resolve species-specific immune responses, host–pathogen interactions, and molecular conservation across species [19,63,65].

Historically, veterinary applications of ST were constrained by technical challenges, such as limited access to species-specific reagents and platform compatibility. However, these barriers are diminishing as transcriptomic atlases increasingly incorporate tissues from dogs, pigs, cattle, and zebrafish [1,31,57,58,59]. Researchers are now leveraging ST to investigate disease mechanisms ranging from infection dynamics to tumor–immune interactions in veterinary contexts [19,57,63].

Notable studies have propelled the field forward. For instance, as previously described, Delorey et al. (2021) constructed a multi-organ ST atlas of SARS-CoV-2 infection, which integrated insights from murine and primate models to chart immune responses across tissue types [68]. In veterinary medicine, a pioneering study applied ST to canine osteosarcoma, offering new insights into tumor microenvironments and disease progression [57]. This application enhances veterinary diagnostics and informs comparative models relevant to zoonotic emergence and pathogen transmission.

Additionally, companion animal research has explored ST in equine neurodegeneration. These applications facilitate the generation of immune–tumor maps that highlight remarkable parallels with human disease biology [19,25,57].

The integration of multi-modal technologies has further strengthened spatial research in veterinary systems. Techniques such as single-cell RNA sequencing, multiplex immunohistochemistry, and in situ hybridization are often combined with ST to validate findings and refine cell-type classification in situ [1,5,6,7]. These synergistic approaches facilitate both diagnostic refinement and translational insight in non-human species.

Despite these advances, the broader application of ST in veterinary medicine remains limited. Public datasets for non-human species are scarce, and many analytical pipelines are optimized for human genomes [3,4,9]. Future directions should prioritize the creation of species-specific spatial atlases, cross-species gene annotation tools, and open-access databases to enhance accessibility and reproducibility. As these resources evolve and as spatial technologies become more cost-effective, ST is increasingly being recognized as a valuable tool for comparative pathology, agricultural innovation, and zoonotic disease investigations—especially given recent studies demonstrating its utility in animal tissues and parasite biology [23]. By leveraging these trends, ST could play a key role in advancing One Health objectives and integrating insights across human, animal, and environmental health.

## 5. Technological Platforms and Analytical Tools

The growth of ST has been driven by rapid advancements in both laboratory platforms and computational tools. These technologies differ in spatial resolution, transcriptome coverage, compatibility with fixed tissues, and ease of integration with complementary methods such as imaging and single-cell sequencing. This section summarizes the key platforms and software used across reviewed studies and highlights how technological choices shape ST applications in health and disease research.

### 5.1. Technologies Used

ST platforms vary in resolution, throughput, and compatibility with FFPE samples. The most widely adopted platform is 10× Genomics Visium, valued for its moderate resolution, FFPE compatibility, and integration with histological imaging [1]. Visium has been commonly used in studies across cancer, brain development, and infectious diseases [5,8,19,25,43,63,68,70,71].

Slide-seq and Slide-seqV2 offer near-single-cell resolution by capturing transcripts using DNA-barcoded beads [15,53]. Recently, Russell (2024) advanced this further by achieving sub-cellular resolution via spatially barcoded single-nucleus profiling [72]. These platforms have been used to map immune cells and zonation in tissues such as the brain and liver [44,46]. Despite their high resolution, adoption remains limited due to technical complexity and reproducibility challenges [3,4].

Ultra-high-resolution platforms like Seq-Scope and Stereo-seq allow subcellular mapping and are particularly useful in oncology and developmental biology [5,6,51,54].

MERFISH and seqFISH use multiplexed fluorescence in situ hybridization (FISH) to detect hundreds to thousands of transcripts per cell [16,73]. These are frequently used in brain and atlas projects where single-molecule resolution is needed. MERFISH, for instance, has enabled spatial mapping of cell types in the mouse brain [25,44,46].

Other platforms like SpaTIAL-seq, Cartana, and Tomo-Seq are applied in specialized contexts such as 3D reconstruction or cleared tissues. Emerging multimodal platforms enable simultaneous profiling of RNA and proteins, useful for immune-oncology and neurodegeneration research [18].

As spatial data become more complex, tools for integration and analysis are equally critical. Platforms are often chosen not just for resolution but for compatibility with imaging workflows, computational pipelines, and FFPE specimens. The distribution of ST platforms across the included literature is visualized in Figure 9.

### 5.2. Tools and Methods Used for Analysis

ST analysis often adapts single-cell RNA-seq workflows to preserve spatial context, requiring integration of specialized tools tailored to both molecular data and tissue architecture. Analysis software is often combined across primary purposes to achieve full spatial data analysis. The most common analysis tools aligned with primary purpose are included in Table 2. A description of many included analysis tools follows.

Seurat is the most widely used spatial analysis tool, providing data integration, clustering, dimensionality reduction, and spatial mapping capabilities [2,5,8,14,28]. Scanpy is a Python-based alternative providing similar functions with greater flexibility for large datasets. Squidpy, an extension of Scanpy, specializes in spatial graph modeling and image integration [27]. These Python-based applications provide specialized single-cell spatial analysis and graph modeling within the Python environment.

CIBERSORT is used for cell-type deconvolution, while Harmony is primarily applied for batch effect correction in integrated spatial and single-cell datasets [10,18,48]. Analytical methods such as Leiden clustering, principal component analysis, and uniform manifold approximation and projection are used to identify spatial domains and visualize transcriptional gradients [5,6,27]. Monocle reconstructs cellular trajectories and infers pseudotime in spatial contexts [2,14]. A summary of tools and their use cases is provided in Table 2. The frequency of software tool usage across ST publications is summarized in Figure 10.

### 5.3. Technology Adoption over Time

The evolution of ST platforms reflects a transformative shift from early custom-built tools to widely accessible commercial technologies. A key milestone came in 2016, with the introduction of the first barcoded array-based spatial system—a foundational approach that led to the development of 10× Genomics Visium, which significantly lowered the technical barriers to entry and catalyzed widespread adoption of ST in biomedical research [1,3,4,9].

Subsequently, high-resolution platforms such as Slide-seqV2, MERFISH, and CosMx Spatial Molecular Imager have emerged, offering specialized capabilities for subcellular resolution, high transcript throughput, and tissue-specific profiling—especially in oncology, neuroscience, and immune tissue studies [15,16,18,53,73]. However, unlike Visium, these platforms require advanced imaging systems, custom reagent design, and computational workflows optimized for human genomes, which continue to pose adoption challenges. As a result, high-resolution ST remains underutilized, particularly in non-model organisms and veterinary research, thereby limiting direct cross-species comparisons.

To overcome these limitations, researchers frequently integrate ST with complementary methods such as single-cell RNA sequencing (scRNA-seq), immunohistochemistry, and in situ hybridization. For example, Khaliq et al. (2024) combined spatial and single-cell profiling to resolve tumor subclones and their microenvironmental context [66], while Stur et al. (2022) employed immune deconvolution alongside ST to map cellular immune dynamics in ovarian cancer [74].

These multi-modal approaches provide a more comprehensive understanding of tissue structure and disease mechanisms by capturing both spatial and molecular complexity. As ST technologies become increasingly modular, and as analytical pipelines become more standardized and species-adaptable, ST is expected to play a central role in diagnostic innovation, personalized therapeutics, and comparative biomedical research.

## 6. Discussion

This scoping review maps the rapidly growing field of ST across quantitative and thematic elements in human and veterinary medicine applications. From an analysis of 1398 studies published between 2016 and early 2025, we identified key research focuses, platform trends, popular analysis tools, cross-species applications, and underexplored research domains. Our findings highlight emerging maturation of the field, translating ST from a highly specialized technology to an accessible, widely adopted tool for studying gene expression in tissue microenvironments across biological systems. However, persistent gaps exist within the promise, including limited use in diverse diseases and non-human species, and there are also challenges in data standardization and reproducibility. This review solidifies ST as a foundational method for studying gene expression analysis to improve abilities to diagnose, treat, and prevent disease through biomedical and veterinary research.

### 6.1. Summary of Key Findings

Cancer remains the dominant focus of ST research, particularly in human tumors of the breast, liver, lung, gastrointestinal tract, and brain. The study of cancer directly benefits from ST’s ability to capture spatially localized gene expression, map immune cell infiltration, and resolve tumor heterogeneity through employment of specific platforms and analysis tools. These applications address major clinical needs by enabling the identification of therapy-resistant niches, characterization of stromal–tumor interactions, and delineation of cellular gradients—insights that were previously difficult to achieve using bulk or single-cell methods due to their lack of spatial resolution.

At the same time, significant technological advancements have shaped the field’s development. 10× Genomics Visium emerged as the most widely adopted platform due to its accessibility, FFPE compatibility, and ease of use. Although it is not considered a high-resolution platform, its user-friendly design has removed many of the barriers associated with more specialized spatial methods, facilitating broader adoption across both research and translational settings. In parallel, there is increasing use of high-resolution platforms like Slide-seq, MERFISH, and Stereo-seq in contexts requiring single-cell or subcellular granularity, typically representing specialty research disciplines such as cancer and developmental biology.

An important technological advancement in ST is its integration with other omics modalities, such as single-cell RNA sequencing (scRNA-seq), proteomics, and imaging. These multi-modal approaches enable a more comprehensive understanding of complex biological systems, including tumor immunology, neurodevelopment, and inflammation. In practice, many ST studies rely on scRNA-seq for accurate cell-type annotation and on proteomics for validating functional insights. This interdependence highlights the interdisciplinary nature of ST and its growing role within modern systems biology, where combining multiple data types is key to unraveling intricate cellular and molecular biology.

In addition to the diagnostic, therapeutic, and mechanistic application types summarized in the results, our thematic synthesis identified two recurring patterns across the literature: an increasing use of multi-modal ST strategies and a growing emphasis on spatial resolution. These trends appeared across several research domains, including but not limited to oncology, neurobiology, and immunology, reflecting broader technological and translational shifts in the field.

Notably, early ST studies were primarily descriptive or discovery-driven, focusing on exploratory mapping of tissues and construction of atlases. However, recent years have seen a transition toward more hypothesis-driven and application-specific research, particularly in oncology, regenerative medicine, and infectious diseases. This shift signals a growing maturity of the field, with ST increasingly employed in clinically relevant investigations aimed at biomarker validation, treatment stratification, and disease modeling.

While technological and methodological advancements continue to drive the field forward, a persistent imbalance in species representation is evident. Human and murine models dominate current ST research, whereas species such as dogs, pigs, zebrafish, and non-human primates remain underutilized. Expanding research in these models is essential not only for understanding species-specific biology but also for advancing comparative and translational applications. Such efforts are critical to achieving the goals of One Health by fostering integrative insights across human, animal, and environmental health domains. Broadening species diversity in ST will ultimately enhance the clinical relevance and global applicability of this technology.

Finally, the journal landscape for ST research has evolved considerably. Early publications were predominantly featured in high-impact journals during the technology’s initial breakthroughs. Over time, ST has also gained presence in specialized journals across fields such as neuroscience, oncology, and computational biology. This diversification reflects the growing relevance of spatial technologies in both foundational and translational research and suggests increasing recognition of ST’s potential across scientific disciplines.

### 6.2. Cross-Species and Comparative Insights

Veterinary and comparative pathology research are underrepresented in the ST literature. Critical studies in mice, fish, livestock, and non-human primates have mapped immune responses, modeled infection, and revealed cellular insights within tumor microenvironments. However, murine models dominate the comparative pathology landscape, and veterinary insights are very limited across all model species.

Cross-species transcriptomic comparisons offer valuable perspectives into both conserved biological processes and species-specific differences that can enhance translational modeling. For example, Hodge et al. (2019) compared the cerebral cortex of humans and mice and found that while core neuronal and metabolic pathways were conserved, immune-related gene expression and glial cell profiles showed species-specific divergence [75]. In a related study, Qu et al. (2022) constructed a single-cell transcriptomic and regulomic atlas of cynomolgus monkeys and compared it with human and mouse data, identifying core regulatory circuits conserved across species, such as metabolic and developmental transcription factors, alongside divergent immune signaling and epithelial differentiation pathways [76]. These findings support the application of ST to veterinary species for evaluating disease mechanisms that may align with or differ from those in humans. Such comparative insight strengthens the translational relevance of veterinary models, enabling identification of shared diagnostic targets and species-specific therapeutic opportunities, particularly within the One Health framework.

Crucially, spatial profiling of immune responses at mucosal surfaces—where zoonotic pathogens first gain entry—directly addresses surveillance needs. Dai et al. (2023) performed scRNA-seq on chicken lung tissue infected with H5N1 Avian Influenza, identifying inflammatory macrophages and epithelial cells as primary sites of viral replication and cytokine release [77]. While not using ST, this study defines precise cellular targets within the respiratory interface where spatial mapping would be most impactful. More directly, Levinger et al. (2025) employed spatial and single-cell transcriptomics in the Egyptian fruit bat to map immune gene expression across gut and lung epithelial layers [78]. They uncovered elevated complement gene expression within discrete mucosal zones implicated in barrier defense and viral tolerance. This is among the first demonstrations of ST to localize immune adaptations at host–environment interfaces in a reservoir species.

These examples illustrate how ST can delineate the in situ spatial architecture of immune responses at the precise tissue sites where zoonotic spillover may occur. Incorporating ST into veterinary surveillance models will enhance our ability to identify early infection foci, understand species-specific immune compartments, and strengthen One Health approaches for emerging infectious disease monitoring.

### 6.3. Technology and Analytical Tools

ST technology selection significantly shapes research capabilities and outcomes including resolution, data exports, and multi modal integration opportunities. Visium is the most commonly used platform in the literature due to its accessibility, originating from being one of the earliest commercially available platforms. Visium removed many of the technological barriers to enter the field of ST and encouraged diverse applications across fields and research questions.

Advanced, high-resolution tools including MERFISH, Slide-seq, and Stereo-seq are increasing in popularity for specific research disciplines including developmental biology and oncology due to their higher-resolution capabilities for studying microenvironments. The study of developmental biology and oncology requires precise spatial mapping at cellular and subcellular resolutions to identify cellular interactions and developmental processes at a fine scale. However, currently, specialized infrastructure limits widescale accessibility and adoption, as seen with Visium.

Analytical tools including Seurat, Scanpy, Squidpy, and CIBERSORT are commonly used to visualize spatial transcriptomic data for enhanced identification and understanding of specific gene expression patterns and inter-tissue spatial domains and to perform cell-type deconvolution. Despite the capabilities of current analytical tools and rapid development of new tools, differences in preprocessing, normalization, and spatial resolution pose challenges for scientific reproducibility and cross-study comparisons. Therefore, future development within the field should focus on establishing standardized pipelines and ensuring transparent metadata reporting mechanisms to enhance data consistency and cross-species comparison opportunities across ST research.

### 6.4. Research Gaps and Limitations of the Literature

While this review provides a broad overview of ST applications in human and veterinary medicine, several limitations must be acknowledged. First, methodological inconsistency across studies limits reproducibility and cross-study comparisons. Many publications do not consistently report key parameters such as spatial resolution, tissue preservation protocols, data normalization techniques, or computational analysis pipelines. This variability complicates the interpretation of findings and impedes standardization across the field. Second, species representation remains heavily skewed toward human and mouse models. Research in livestock, companion animals, and wildlife species is notably sparse. One major barrier to expanding ST in veterinary and agricultural contexts is the challenge of RNA degradation during tissue collection and processing. For example, farm-collected or field-sourced samples often experience delayed preservation or inadequate freezing conditions, compromising RNA integrity and the quality of spatial data [79,80].

Third, the cost and technical demands of ST remain prohibitive for many research settings. Compared to single-cell RNA sequencing (scRNA-seq), ST requires higher sequencing depth, specialized platforms (e.g., 10× Genomics Visium), and meticulous tissue handling—all of which increase financial and logistical burden. Although scRNA-seq is more scalable and cost-effective, it inherently lacks spatial resolution, highlighting the trade-off between accessibility and contextual insight [81]. Fourth, there is a clear organ-system bias in the current body of literature. The majority of studies in the ST literature focus on tissues such as the brain, liver, and tumors—likely due to their clinical importance and compatibility with existing ST platforms. In contrast, anatomically and physiologically important tissues like the skin, endocrine glands, and placenta remain underrepresented. This imbalance underscores the need to expand spatial profiling efforts beyond traditional targets, particularly in chronic and reproductive conditions. Additionally, the field is limited by a lack of longitudinal studies tracking spatial gene expression changes over the course of disease. Without such temporal data, it is difficult to understand how ST dynamics evolve during disease progression, which restricts the translational applicability of findings.

To address these gaps, future research should prioritize the standardization of ST methodologies, including tissue preservation protocols, spatial resolution reporting, and data analysis pipelines. Methodological consistency will improve reproducibility and enable better comparison across studies. Simultaneously, targeted efforts are needed to explore underexamined tissues and organ systems, especially in veterinary species, to support the goals of One Health and comparative pathology.

However, recent advances suggest that these gaps are beginning to be addressed in clinical veterinary research. For example, Beck et al. (2025) applied NanoString GeoMx Digital Spatial Profiling to formalin-fixed osteosarcoma tissues from pet dogs enrolled in a clinical trial, identifying spatial gene expression patterns associated with disease-free intervals [82]. This study illustrated the potential of ST to enhance diagnostics and treatment strategies in companion animals, marking a key step toward the adoption of precision veterinary medicine. As ST becomes more integrated into veterinary and translational research, ethical considerations must be proactively addressed. These include informed client consent, animal welfare, and transparent data governance—areas guided by policies from the U.S. Food and Drug Administration (FDA) Center for Veterinary Medicine and the American Veterinary Medical Association (AVMA) [83].

### 6.5. Methodological Limitations of This Review

This study followed PRISMA-ScR guidelines for scoping reviews to systematically map the publication landscape within the field of ST. Limitations of this review include restriction to English-language literature and reliance on database search results. Screening and data extraction were conducted by a single reviewer, which may introduce subjectivity despite structured protocols and adherence to pre-established inclusion and exclusion criteria.

## 7. Conclusions

ST has transformed how researchers study states of health and disease across biological systems by providing positional context to transcriptional data. Novel insights at cellular and subcellular levels enhance our understanding of various biological systems in mechanistic, diagnostic, and therapeutic contexts. Currently, research for human diseases dominates the publication landscape across human and animal models, with a strong focuses on oncology and neurobiology. Fragmented data highlight disparities in the literature across study of diverse diseases and model systems. This highlights critical gaps in current applications of ST and the potential to increase applications across targeted specialized research in veterinary medicine, infectious disease, comparative pathology, and underexplored human conditions. Our review highlights the achievements, gaps, opportunities, and current limitations of this rapidly evolving field from the pioneering publication in 2016 [1] through February 2025.

Future efforts in ST research should focus on broadening species and disease representation, improving data transparency, and supporting methodological standardization. ST has shifted from a specialized methodology to a foundational tool for systems biology, translational medicine, and precision health across mechanistic, diagnostic, and therapeutic applications for varied species and disciplines. ST is providing novel understanding into the structure and function of varied microenvironments in biomedical and veterinary medical research.

## Figures and Tables

**Figure 1 ijms-26-06163-f001:**
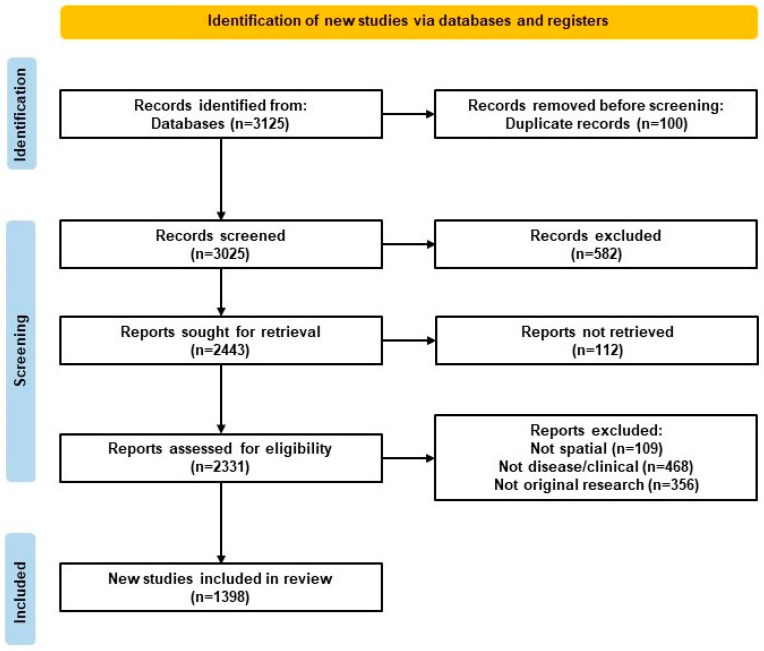
Preferred Reporting Items for Systematic Reviews and Meta-Analyses Extension for Scoping Reviews (PRISMA-ScR) flow diagram. Flow diagram of study selection for the scoping review, following PRISMA-ScR guidelines and including identification, screening, eligibility, inclusion, and reasons for exclusion.

**Figure 2 ijms-26-06163-f002:**
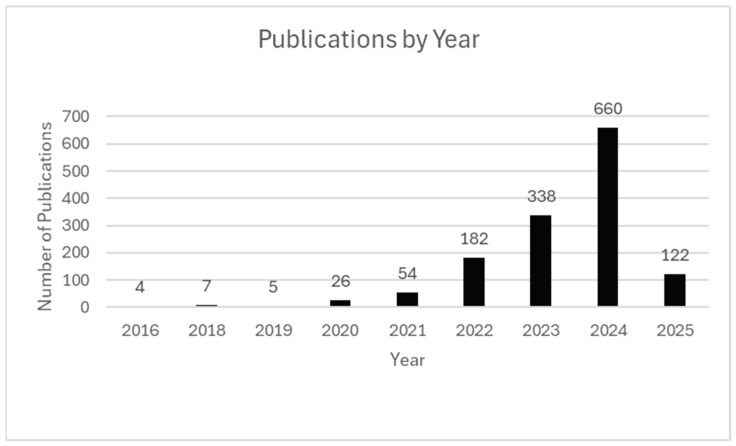
Publications by Year. Annual count of ST publications from 2016 to early 2025. Peaks correspond to major milestones, including the 2016 introduction, commercial release of Visium in 2019, and increased activity during the COVID-19 pandemic. Publication data for 2025 are current through 7 February 2025.

**Figure 3 ijms-26-06163-f003:**
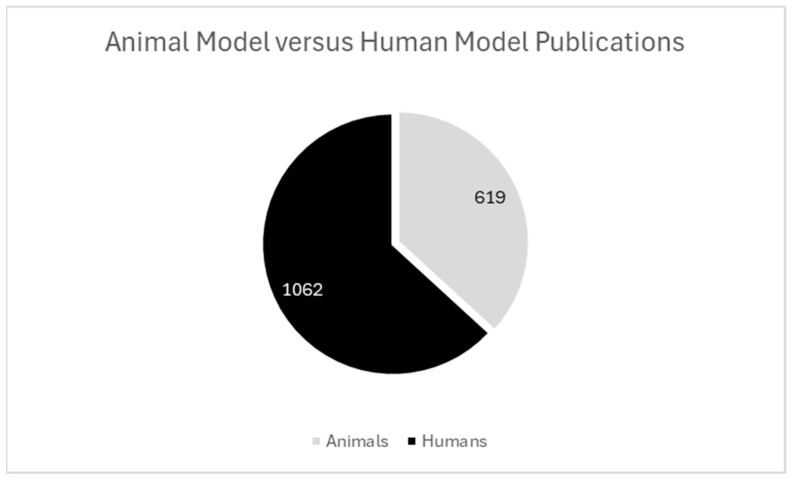
Animal model versus human model publications. Among the studies included, 1062 used human models, and 619 used animal models. Combined models within publications are represented as individual datapoints.

**Figure 4 ijms-26-06163-f004:**
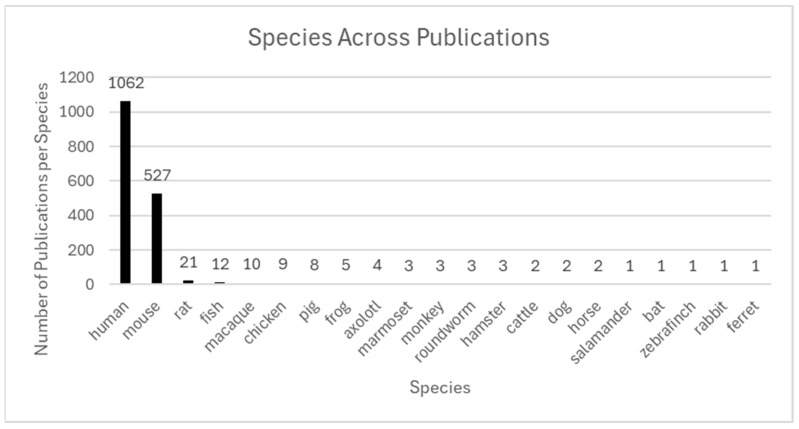
Species across publications. The figure shows the number of publications utilizing ST across various species. While human (*n* = 1062) and mouse (*n* = 527) dominate, a diversity of other models is emerging. Non-human primates (*n* = 21) and rat (*n* = 16) are used in translational and preclinical research. Lower-frequency models reflect specialized biological questions: for example, fish and frog (developmental biology), chicken (avian immunology), pig and cattle (large-animal translational models), axolotl (regenerative medicine), roundworm (aging and development), hamster and ferret (respiratory infectious disease models), bat (zoonotic virus reservoir studies), dog and horse (veterinary and comparative medicine), salamander (regeneration), and rabbit and zebra finch (neuroscience and vocalization). The inclusion of such rare species highlights the expanding utility of ST in comparative and cross-species biology.

**Figure 5 ijms-26-06163-f005:**
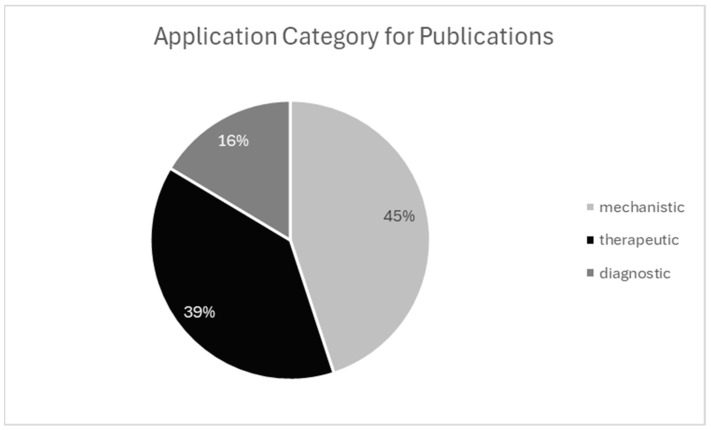
Application category for publication. Categorization of studies by primary research aim. Mechanistic studies were most common, followed by therapeutic and diagnostic.

**Figure 6 ijms-26-06163-f006:**
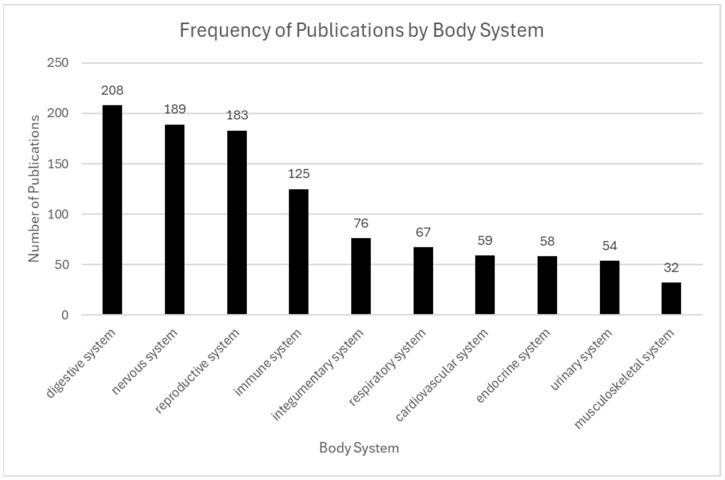
Frequency of publications by body system. The digestive and nervous systems were the most frequently studied, appearing in 208 publications and 189 publications, respectively. The reproductive and immune systems were also prominent, with 183 and 125 publications, respectively. In contrast, systems such as cardiovascular, urinary, integumentary, and musculoskeletal were represented less frequently, each with 60 or fewer publications.

**Figure 7 ijms-26-06163-f007:**
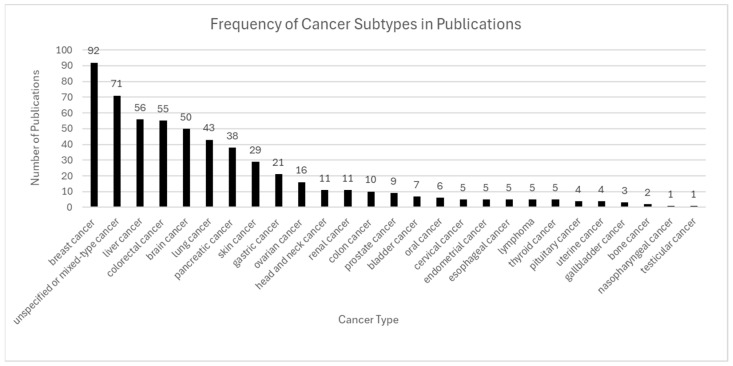
Frequency of cancer subtypes in publications. Breast, liver, colorectal, brain, and lung cancers were the most frequently studied, reflecting their clinical significance and spatial heterogeneity. Other cancer types, such as gastric, head and neck, and prostate cancers, also appeared regularly, while rare cancers like testicular and nasopharyngeal cancer were minimally represented.

**Figure 8 ijms-26-06163-f008:**
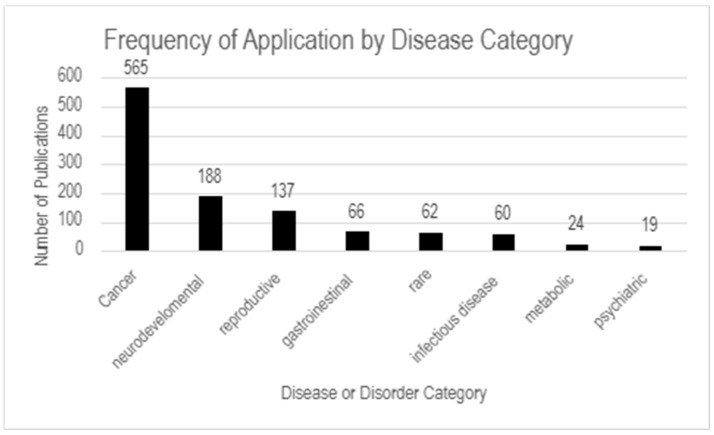
Frequency of application by disease category. The highest number of publications focus on cancer (565), followed by neurodevelopmental (188) and reproductive diseases (137). Less represented categories include gastrointestinal (66), rare diseases (62), infectious diseases (60), metabolic diseases (24), and psychiatric disorders (19).

**Figure 9 ijms-26-06163-f009:**
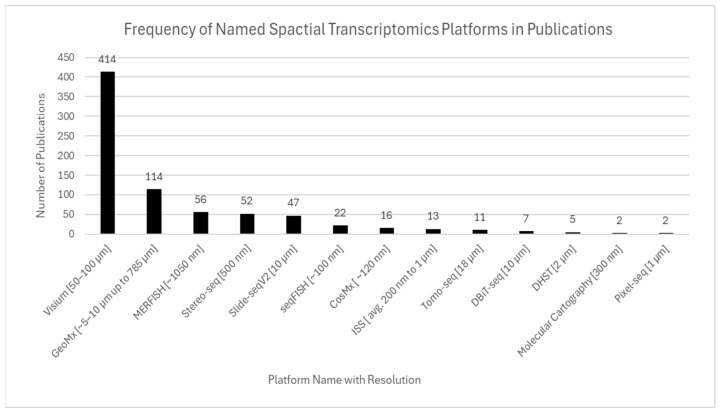
Frequency of named ST platforms in publications. 10× Genomics Visium was the most widely used platform, reflecting its accessibility and compatibility with formalin-fixed tissues. High-resolution and specialized platforms such as Slide-seq, Slide-seqV2, MERFISH, Seq-Scope, and Stereo-seq were also represented, though less frequently. The figure highlights both the dominance of Visium in mainstream research and the growing adoption of alternative technologies tailored for specific resolution and throughput needs.

**Figure 10 ijms-26-06163-f010:**
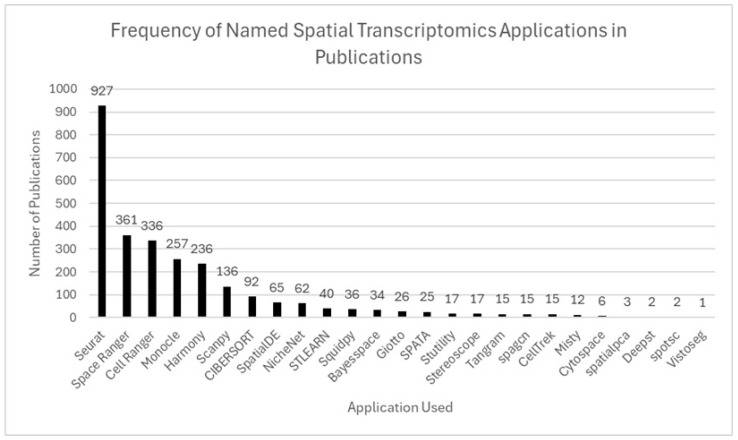
Frequency of software tools used in ST publications. Seurat was the most widely used tool (927 publications), followed by Space Ranger, Cell Ranger, Monocle, and Harmony. Other tools, including Scanpy, SpatialDE, and Giotto, were used less frequently, reflecting both dominant platforms and an expanding ecosystem of specialized applications.

**Table 1 ijms-26-06163-t001:** Inclusion and Exclusion Criteria.

Inclusion Criteria	Exclusion Criteria
Studies conducted in animal or human health contexts	Non-spatial transcriptomics studies (e.g., bulk or scRNA-seq without spatial data)
Peer-reviewed journal articles	Studies outside the scope of human or veterinary health and disease, including those focused on method development without new biological/clinical insights
Original research articles, systematic reviews, or meta-analyses	Non-peer-reviewed sources (e.g., preprints, conference abstracts, and theses)
Published between 2016 and 7 February 2025	Case reports and scoping reviews lacking comparative analysis or methodological rigor
Studies employing animal models, human samples, or publicly available human and animal health datasets	Studies not published in English
Focus on ST techniques	
Report biological or clinically relevant findings	

**Table 2 ijms-26-06163-t002:** Analysis tools used in ST applications.

Primary Purpose	Software Tool	Use Case/Description
Preprocessing and Primary Analysis	Space Ranger	10× Genomics pipeline for alignment, barcode filtering, and quantification Link: https://support.10xgenomics.com/spatial-gene-expression/software/pipelines/latest/installation accessed on 23 June 2025
	STARsolo	Alignment and UMI quantification for single-cell and spatial RNA-seq Link: https://github.com/alexdobin/STAR accessed on 23 June 2025
	Slide-seq Tools	Pipeline for decoding bead barcodes and aligning Slide-seq data Link: https://github.com/MacoskoLab/slideseq-tools accessed on 23 June 2025
	Stereo-seq Pipeline	BGI’s pipeline for high-resolution Stereo-seq data processing Link: https://github.com/BGIResearch/stereopy accessed on 23 June 2025
Single-cell and Spatial Integration	Seurat (v5.0+)	R-based toolkit for scRNA-seq and spatial integration using anchors Link: https://github.com/satijalab/seurat accessed on 23 June 2025
	Scanpy	Python-based toolkit for scalable single-cell analysis Link: https://github.com/scverse/scanpy accessed on 23 June 2025
	Squidpy	Python >=3.9 library for spatial graph-based analyses and neighborhood enrichment Link: https://github.com/scverse/squidpy accessed on 23 June 2025
Differential Expression and Statistics	edgeR	Statistical analysis of differential expression in count data Link: https://bioconductor.org/packages/release/bioc/html/edgeR.html accessed on 23 June 2025
	DESeq2	Model-based differential expression testing for count data Link: https://bioconductor.org/packages/release/bioc/html/DESeq2.html accessed on 23 June 2025
	SpatialDE	Detects spatially variable genes from ST data Link: https://github.com/Teichlab/SpatialDE accessed on 23 June 2025
Spatial Patterns and Neighborhood Analysis	SpaGCN	Graph convolutional network to detect spatial domains and gene patterns Link: https://github.com/jianhuupenn/SpaGCN accessed on 23 June 2025
	BayesSpace	Bayesian clustering for high-resolution spatial domain detection Link: https://github.com/edward130603/BayesSpace accessed on 23 June 2025
	Giotto	Visual analytics platform optimized for large tissue sections and 3D data Link: https://github.com/RubD/Giotto accessed on 23 June 2025
	stLearn	Integrates histology and spatial gene expression for cell–cell interaction inference Link: https://github.com/BiomedicalMachineLearning/stLearn accessed on 23 June 2025
	SpatialExperiment	R/Bioconductor framework for spatial omics data representation and analysis Link: https://bioconductor.org/packages/release/bioc/html/SpatialExperiment.html accessed on 23 June 2025
Cell-Type Deconvolution	CIBERSORT	Deconvolution of immune cell types from bulk or ST data Link: https://cibersort.stanford.edu/ accessed on 23 June 2025
	Harmony	Batch correction and integration across spatial or single-cell datasets Link: https://github.com/immunogenomics/harmony accessed on 23 June 2025
	RCTD	Maps cell types from scRNA-seq onto spatial locations using probabilistic assignment Link: https://github.com/dmcable/RCTD accessed on 23 June 2025
	cell2location	Probabilistic deconvolution of ST maps with scRNA-seq reference Link: https://github.com/BayraktarLab/cell2location accessed on 23 June 2025
	SPOTlight	NMF-based cell-type deconvolution method for ST Link: https://github.com/MarcElosua/SPOTlight accessed on 23 June 2025
	DestVI	Variational inference model for cell-type deconvolution in spatial data Link: https://github.com/YosefLab/scvi-tools accessed on 23 June 2025

## Data Availability

The search strategies and screening tags used in this study are available in an Excel file deposited in Zenodo https://doi.org/10.5281/zenodo.15717108, accessed on 22 June 2025.

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
