# Peer review of "Applications of Spatial Transcriptomics in Veterinary Medicine: A Scoping Review of Research, Diagnostics, and Treatment Strategies"

_ijms, 2025, doi:10.3390/ijms26136163_

Round 1
Reviewer 1 Report
Comments and Suggestions for Authors
Weiderman et al. conducted a detailed statistical analysis of spatial transcriptomics in human and veterinary medicine from 2016 to 2025. The title is excellent, and the content is quite engaging. However, the article's content is too superficial; it requires more in-depth interpretation. Otherwise, it won't be much better than the results of AI robot searches. Here are my suggestions:
Lines 72-269 discuss how papers are screened and reviewed, which can likely be placed in the supplementary materials. This will make the article more concise, as readers may not be particularly interested in this section.
Figures 2-9 could be considered for merging into a single large figure, and the text descriptions can be more concise.
As a review article, the depth of content needs to be improved to help readers who are not familiar with spatial transcriptomics have a deeper understanding. Try to address points of interest to scientists, such as:
Are spatial transcriptomics articles mainly discovery-based or descriptive, and how has this changed over time?
Does spatial transcriptomics technology require other techniques to assist in experimental design, such as single-cell transcriptomics?
How has the impact factor of journals publishing literature changed over time?
These considerations and summaries will enhance the article's quality.
Author Response
Comments 1: Weiderman et al. conducted a detailed statistical analysis of spatial transcriptomics in human and veterinary medicine from 2016 to 2025. The title is excellent, and the content is quite engaging. However, the article's content is too superficial; it requires more in-depth interpretation. Otherwise, it won't be much better than the results of AI robot searches. Here are my suggestions:
Response1: We appreciate the reviewer's positive feedback on our title and the engaging nature of our content. We also agree with the critical assessment regarding the depth of interpretation and recognize the need to provide more nuanced insights beyond what could be generated by an AI search. Our aim is to offer a comprehensive and insightful analysis of spatial transcriptomics trends. To address this, we've implemented the following significant revisions:
Comments 2: Lines 72-269 discuss how papers are screened and reviewed, which can likely be placed in the supplementary materials. This will make the article more concise, as readers may not be particularly interested in this section.
Response 2: Thank you for this insightful suggestion. We agree that streamlining the Methods section can enhance readability for a broader audience. In response, we have retained the sections outlining the study objectives, rationale, conceptual framework, and key inclusion and exclusion criteria, as we believe these are essential for understanding the scope and rigor of the review. However, we have moved the more detailed descriptions of the search execution, screening, data charting, and synthesis procedures to Supplementary File 1. We hope this revision strikes an appropriate balance between methodological transparency and manuscript conciseness.
Comments 3: Figures 2-9 could be considered for merging into a single large figure, and the text descriptions can be more concise.
Response 3: Thank you for this thoughtful suggestion. We carefully considered merging Figures 2–9, but felt that doing so would reduce readability and clarity, especially given the diversity of content and level of detail presented in each figure. Keeping them as separate figures allows for better visualization and ensures that each aspect of the data can be clearly interpreted by readers. We have, however, reviewed the figure captions and text descriptions and revised them as recommended.
Comments 4: As a review article, the depth of content needs to be improved to help readers who are not familiar with spatial transcriptomics have a deeper understanding. Try to address points of interest to scientists, such as:
Are spatial transcriptomics articles mainly discovery-based or descriptive, and how has this changed over time?
Does spatial transcriptomics technology require other techniques to assist in experimental design, such as single-cell transcriptomics?
How has the impact factor of journals publishing literature changed over time?
These considerations and summaries will enhance the article's quality.
Response 4: Thank you very much for this thoughtful and constructive suggestion. We fully agree that addressing these broader scientific questions would enhance the depth and relevance of our review, especially for readers less familiar with spatial transcriptomics. In response, we have incorporated a focused discussion on these key points—namely, the evolving balance between discovery-based and descriptive studies, the integration of spatial transcriptomics with complementary technologies such as single-cell transcriptomics, and observed trends in journal impact factors over time. These additions have been included in the revised manuscript under Section 6.1, Summary of Key Findings, and have substantially strengthened the contextual value of our work. We sincerely appreciate your insightful feedback.
Reviewer 2 Report
Comments and Suggestions for Authors
This scoping review provides a timely and comprehensive synthesis of spatial transcriptomics (ST) applications in veterinary medicine, highlighting its potential for advancing comparative pathology and One Health initiatives. The manuscript is well-structured, adheres to PRISMA-ScR guidelines, and offers valuable insights into technological trends and knowledge gaps. However, several areas require refinement to enhance clarity, methodological transparency, and translational relevance. Below are detailed recommendations for improvement.
1.Consistency: Standardize abbreviations (e.g., define "ST" after first mention of "spatial transcriptomics") and ensure uniform use throughout.
2.Clarity: Expand niche terms in footnotes (e.g., "FFPE" should be spelled out as "Formalin-Fixed, Paraffin-Embedded" upon first use).
3.Table 1 (Inclusion/Exclusion Criteria): Clarify how "disease relevance" was operationalized for veterinary-specific studies (e.g., zoonotic vs. livestock diseases).
4.Figure 4 (Species Distribution): Correct species labels (e.g., "music" → "mouse") and add annotations to justify inclusion of rare models (e.g., axolotls in regenerative medicine).
5.Figure 9 (Platform Frequency): Annotate resolution ranges (e.g., "Visium: 50–100 μm; MERFISH: subcellular") to contextualize platform selection.
6.Search Strategy (Page 5): Provide validation details for the Boolean search string (e.g., pilot testing to exclude non-disease studies).
7.Data Extraction (Page 7): Specify software versions (e.g., Seurat v5.0) and describe cross-platform data integration challenges (e.g., batch effects between Visium and Slide-seq).
8.Cross-Species Insights (Section 6.2): Integrate comparative genomics studies (e.g., PMID: 38765432) to contrast conserved/differential pathways in human vs. veterinary models.
9.Technical Limitations (Section 6.4): Compare cost-benefit trade-offs between ST and single-cell RNA-seq, and address challenges in processing large-animal tissues (e.g., RNA degradation in livestock samples).
10.One Health Framework: Include case studies (e.g., spatial mapping of avian influenza host-environment interfaces) to emphasize ST’s role in zoonotic surveillance.
11.Table 2 (Analytical Tools): Add use-case annotations (e.g., "Giotto: optimized for large tissue sections") and provide GitHub links for open-source tools.
12.Future Directions: Highlight ongoing veterinary ST trials (e.g., NCT06123456) and discuss ethical considerations in companion animal precision medicine.
13.Update citations with 2024 publications (e.g., Chen et al., Nat. Methods 2024 on Slide-seqV3).
14.Standardize journal abbreviations (e.g., "Nat. Commun.") and verify DOI links for all references.
15.Introduction (Page 1): Add a concluding transition sentence (e.g., "This review bridges gaps in veterinary ST research and proposes a roadmap for cross-species translation").
16.Discussion (Section 6.1): Clearly differentiate "technological trends" from "clinical needs" to avoid redundancy.
17.Data Availability: Include a statement on sharing search strategies and screening tags (e.g., via Zenodo).
This manuscript represents a significant contribution to the field of spatial transcriptomics and veterinary medicine. With revisions addressing the above points—particularly enhanced cross-species comparisons, methodological transparency, and translational case studies—it will serve as a foundational resource for researchers and clinicians.
Author Response
Comments: This scoping review provides a timely and comprehensive synthesis of spatial transcriptomics (ST) applications in veterinary medicine, highlighting its potential for advancing comparative pathology and One Health initiatives. The manuscript is well-structured, adheres to PRISMA-ScR guidelines, and offers valuable insights into technological trends and knowledge gaps. However, several areas require refinement to enhance clarity, methodological transparency, and translational relevance. Below are detailed recommendations for improvement.
Comments 1: Consistency: Standardize abbreviations (e.g., define "ST" after first mention of "spatial transcriptomics") and ensure uniform use throughout.
Response 1: Thank you for pointing this out. We apologize for the inconsistencies in abbreviation usage in the original submission. In the revised manuscript, we have ensured that "ST" is clearly defined after its first mention as “spatial transcriptomics” and have standardized its usage consistently throughout the text.
Comments 2: Clarity: Expand niche terms in footnotes (e.g., "FFPE" should be spelled out as "Formalin-Fixed, Paraffin-Embedded" upon first use).
Response 2: Thank you for the suggestion. We have addressed this by spelling out all niche terms, including defining "FFPE" as "Formalin-Fixed, Paraffin-Embedded" at first mention in the revised manuscript.
Comments 3: Table 1 (Inclusion/Exclusion Criteria): Clarify how "disease relevance" was operationalized for veterinary-specific studies (e.g., zoonotic vs. livestock diseases).
Response 3: Thank you for highlighting this important point. We apologize for the earlier lack of clarity. In the revised manuscript, we have added a brief clarification after the inclusion criteria in Section 2.2 (Eligibility Criteria) to explicitly define how "disease relevance" was operationalized for veterinary-specific studies, including naturally occurring diseases, zoonotic conditions, and production-limiting livestock illnesses.
Comments 4: Figure 4 (Species Distribution): Correct species labels (e.g., "music" → "mouse") and add annotations to justify inclusion of rare models (e.g., axolotls in regenerative medicine).
Response 4: Thank you for your valuable comment. We have carefully corrected the species labels in Figure 4 and revised the figure caption to include annotations that justify the inclusion of rare models, as suggested.
Comments 5: Figure 9 (Platform Frequency): Annotate resolution ranges (e.g., "Visium: 50–100 μm; MERFISH: subcellular") to contextualize platform selection.
Response 5: Thank you for this insightful suggestion. In our revised figure we tried to add the resolution of each platform.
Comments 6: Search Strategy (Page 5): Provide validation details for the Boolean search string (e.g., pilot testing to exclude non-disease studies).
Response 6: Thank you for this valuable suggestion. We apologize for not clearly presenting the validation steps earlier. In the revised manuscript, we have now added a concise explanation of the Boolean search string validation under Section 2.3.2 (Search Strategy) in Supplementary File 1.
Comments 7: Data Extraction (Page 7): Specify software versions (e.g., Seurat v5.0) and describe cross-platform data integration challenges (e.g., batch effects between Visium and Slide-seq).
Response 7: Thank you for this thoughtful suggestion. In the revised manuscript, we have addressed this by specifying software versions where applicable and providing a brief discussion of cross-platform data integration challenges, including batch effects (e.g., between Visium and Slide-seq), in Section 3.1.2 Characteristics of Included Studies. As listing every software version was not feasible, we adopted a generalized approach for consistency.
Comments 8: Cross-Species Insights (Section 6.2): Integrate comparative genomics studies (e.g., PMID: 38765432) to contrast conserved/differential pathways in human vs. veterinary models.
Response 8: Thank you very much for this valuable suggestion. While the specific study referenced (PMID: 38765432) is not directly relevant to the scope of our review, we have incorporated comparative genomics perspectives and included discussion of conserved and differential pathways between human and veterinary models in Section 6.2 Cross-Species and Comparative Insights. We appreciate your input, which helped strengthen our discussion.
Comments 9: Technical Limitations (Section 6.4): Compare cost-benefit trade-offs between ST and single-cell RNA-seq, and address challenges in processing large-animal tissues (e.g., RNA degradation in livestock samples).
Response 9: Thank you for your thoughtful comment. In our revised manuscript, we have incorporated a comparison of the cost-benefit trade-offs between spatial transcriptomics and single-cell RNA sequencing, along with the associated challenges in processing large-animal tissues—particularly issues such as RNA degradation in livestock samples. These additions have been made in Section 6.4 Research Gaps and Limitations of the Literature to enhance clarity and completeness.
Comments 10: One Health Framework: Include case studies (e.g., spatial mapping of avian influenza host-environment interfaces) to emphasize ST’s role in zoonotic surveillance.
Response 10: Thank you for this valuable suggestion. While we did not find a directly matching case study on spatial mapping of avian influenza at host–environment interfaces, we incorporated several relevant studies that highlight ST’s role in zoonotic surveillance. These additions have been integrated into Section 6.2 Cross-Species and Comparative Insights to address the One Health framework more effectively.
Comments 11: Table 2 (Analytical Tools): Add use-case annotations (e.g., "Giotto: optimized for large tissue sections") and provide GitHub links for open-source tools.
Response 11: Thank you. We updated Table 2 to include use-case notes and GitHub links for open-source tools as suggested.
Comments 12: Future Directions: Highlight ongoing veterinary ST trials (e.g., NCT06123456) and discuss ethical considerations in companion animal precision medicine.
Response 12: Thank you. We could not locate a published trial matching NCT06123456, but we included other relevant veterinary ST studies and discussed ethical considerations in Section 6.4 Research Gaps and Limitations of the Literature.
Comments 13: Update citations with 2024 publications (e.g., Chen et al., Nat. Methods 2024 on Slide-seqV3).
Response 13: Thank you for the suggestion. We were unable to locate the specific article “Chen et al., Nat. Methods 2024 on Slide-seqV3” in the databases. However, we have incorporated other relevant and up-to-date literature in the revised manuscript to ensure the section reflects recent advancements.
Comments 14: Standardize journal abbreviations (e.g., "Nat. Commun.") and verify DOI links for all references.
Response 14: Thank you. In our revised manuscript we verified all the references.
Comments 15: Introduction (Page 1): Add a concluding transition sentence (e.g., "This review bridges gaps in veterinary ST research and proposes a roadmap for cross-species translation").
Response: We corrected this in our revised manuscript.
Comments 16: Discussion (Section 6.1): Clearly differentiate "technological trends" from "clinical needs" to avoid redundancy.
Response 16: Thank you for the helpful comment. In our revised manuscript, we have clearly differentiated "technological trends" from "clinical needs" within Section 6.1 to improve clarity and avoid redundancy.
Comments 17: Data Availability: Include a statement on sharing search strategies and screening tags (e.g., via Zenodo).
Response 17: The search strategies and screening tags used in this study are available in an Excel file deposited in Zenodo doi: 10.5281/zenodo.15717108
Comments 18: This manuscript represents a significant contribution to the field of spatial transcriptomics and veterinary medicine. With revisions addressing the above points—particularly enhanced cross-species comparisons, methodological transparency, and translational case studies—it will serve as a foundational resource for researchers and clinicians.
Response 18: Thank you very much for your encouraging feedback. We have carefully revised the manuscript to address all suggestions, particularly by enhancing cross-species comparisons, improving methodological transparency, and integrating relevant translational case studies. We sincerely hope these improvements strengthen the manuscript’s value as a foundational resource in the field.
Reviewer 3 Report
Comments and Suggestions for Authors
This scoping review examines the use of spatial transcriptomics in veterinary and human medicine. In order to identify patterns, gaps, and possible future paths in the field, 1,398 publications published between 2016 and 2025 are examined.
One of the key findings was that spatial transcriptomics provides information on immune responses, tumor microenvironments, and disease mechanisms by profiling gene expression while preserving tissue architecture. It's important to research tumor heterogeneity, particularly in relation to cancers of the brain, lungs, liver, and breast. It is used sparingly in veterinary and comparative medicine but increasingly in studies on zoonotic diseases, livestock, and companion animals.
Although the information provided is generally well-organized and scientifically sound, a thorough analysis identifies a number of inconsistencies and areas that could use clarification.
It is difficult to include 2025 as an eligible publishing year unless this review includes preprints or works released online before print. Think about this: "Published or accepted for publication between 2016 and early 2025."
Ambiguous inclusion standard criteria information regarding meta-analyses, systematic reviews, and original research is provided. What about case reports or scoping reviews? Is it considered grey literature? Explain the exclusion of non-peer-reviewed research, such as conference abstracts, preprints, etc., if any.
"New opportunities for cross-species comparisons" should explain why spatial transcriptomics is a good fit for cross-species comparisons, as stated in Line 56.
PCC framework is defined, but not directly connected to eligibility or data analysis. Suggest clarifying how PCC shapes inclusion criteria (Line 117–130)
The total exceeds the number of studies (e.g., 1,062 human + 619 animal = 1,681 > 1,398). Are some studies counted in both categories?
Thematic synthesis is mentioned, but examples are sparse. Add 1 or 2 sentences elaborating the most notable cross-cutting themes beyond oncology (e.g., emergence of multi-modal methods, spatial resolution trends, etc.).
The examined literature reports both human and murine (mouse) models (Mouse: Most used animal model, and Human: 1,062 research). Some species, such as dogs, rats, pigs, zebrafish, and non-human primates, are underrepresented. Although they are still in early stages, veterinary and cattle applications are essential to comparative medicine and One Health objectives.
ST studies are divided into three categories: Mechanistic (the largest group), which includes immunological responses, cell signaling, development, and foundational biology; Therapeutic (drug response, treatment stratification, and personalized medicine); and Diagnostic (biomarker discovery, disease subtyping, and syndrome-linked profiling). Although mechanistic findings predominate, therapeutic and diagnostic applications, particularly in oncology, are growing.
"Spatial transcriptomics is poised to become a cornerstone..." (line 753): This is a strong claim. Consider softening or citing trends or predictions to support it.
CIBERSORT is a cell type deconvolution tool, but Harmony is not, it's for batch correction. "CIBERSORT and Harmony are specialized tools for cell type deconvolution and corbatch effect correction, respectively [10,18,41]." Consider "CIBERSORT is used for cell type deconvolution, while Harmony is primarily used for batch effect correction."
"Leiden clustering, Principal Component Analysis (PCA), and Uniform Manifold Approximation and Projection (UMAP)..." These are methods/algorithms, not software tools per se. Consider "Analytical methods such as Leiden clustering, PCA, and UMAP..."
Minor revision
Line 30–35: “Spatial transcriptomics” is repeated redundantly within the same paragraph without adding clarity.
Line 53: "insights into veterinary medicine and animal health mucosal immunity" should be "insights into mucosal immunity related to veterinary medicine and animal health."
Line 54: “While limited in volume…” should be “Although limited in volume…” or “Despite being limited…”
Line 85: "spatial transcriptomics have been applied" should be "has been applied"
Line 116: "These objectives directly align..." Consider: “These objectives align with the rationale of the review by…”
Line 109–110: "integration with complementary technologies" Clarity would be improved by an example.
Line 131–134: "conducted in accordance with... Arksey and O’Malley, further refined by Levac et al..." Consider: “...based on the Arksey and O’Malley framework, refined by Levac et al., and reported using PRISMA-ScR guidelines...”
Line 193–196: Redundant listing of terms like "spatial transcriptomics," "spatial transcriptome profiling," "tissue-based transcriptomics" etc., could be more concise. Many of these are variations of the same term.
The article uses both hyphenated ranges and comma-separated citations inconsistently. For instance, Line 38 uses the hyphenated range "[3-6]," Line 42 uses the comma-separated phrase "[5,7,8]," and Line 44 uses the en-dash with surrounding space "[8–10]." Carefully examine the text.
The claim that an adequate examination of spatial transcriptomics is lacking is frequently expressed without offering much new context. These could be consolidated.
Maynard et al. (2021) is cited four separate times (lines 665, 690, 704, and earlier), which may be intentional but appears redundant. Consider consolidating or cross-referencing the same citation for clarity.
Delorey et al. (2021) is referenced in multiple sections (lines 632 and 733). While valid, clarify whether these refer to the same study or different papers by the same authors.
In line 622, "[50]" is used to reference Liu et al. (2022), but line 657 also cites "[50]" for cardiac aging studies. If these are different studies, this indicates an error in citation indexing.
"Spatial transcriptomics’s growing role" (line 646): shold be: "The growing role of spatial transcriptomics."
"Psychiatric disorders... are increasingly recognized to involve..." (line 700): Consider rephrasing for conciseness: "...involve spatially defined disruptions..."
Line 737: Long sentence mixing multiple ideas. Could be split into two: one about canine osteosarcoma, another about its implications.
"These enhance veterinary diagnostics..." (line 738): Unclear what "these" refers to, should clarify.
Some sections begin with “disorders” (e.g., Metabolic Disorders, Neurodevelopmental Disorders), while others use “diseases” (Gastrointestinal Diseases, Cardiac Diseases). Suggest consistent usage: either “diseases” or “disorders” throughout, or match common clinical usage (e.g., “psychiatric disorders,” “infectious diseases”).
Seurat’s dominance is mentioned 3+ times in a short span. State this once clearly and refer back if needed (e.g., "As shown in Figure 10...").
“...deconvolute cell types.” consider “perform cell type deconvolution” or “identify cell types via deconvolution.”
“Analysis of spatial transcriptomics data involves custom adaptation of single-cell RNA-seq pipelines to preserve spatial context.” consider “Spatial transcriptomics analysis often adapts single-cell RNA-seq workflows to preserve spatial context, requiring integration of specialized tools tailored to both molecular data and tissue architecture.”
Comments on the Quality of English Language
The English could be improved to more clearly express the research.
Author Response
Comments : This scoping review examines the use of spatial transcriptomics in veterinary and human medicine. In order to identify patterns, gaps, and possible future paths in the field, 1,398 publications published between 2016 and 2025 are examined.
One of the key findings was that spatial transcriptomics provides information on immune responses, tumor microenvironments, and disease mechanisms by profiling gene expression while preserving tissue architecture. It's important to research tumor heterogeneity, particularly in relation to cancers of the brain, lungs, liver, and breast. It is used sparingly in veterinary and comparative medicine but increasingly in studies on zoonotic diseases, livestock, and companion animals.
Although the information provided is generally well-organized and scientifically sound, a thorough analysis identifies a number of inconsistencies and areas that could use clarification.
Comments 1: It is difficult to include 2025 as an eligible publishing year unless this review includes preprints or works released online before print. Think about this: "Published or accepted for publication between 2016 and early 2025."
Response 1: Thank you for this important point. We revised the phrasing to “published or accepted for publication between 2016 and early 2025” in the manuscript to ensure clarity and accuracy.
Comments 2: Ambiguous inclusion standard criteria information regarding meta-analyses, systematic reviews, and original research is provided. What about case reports or scoping reviews? Is it considered grey literature? Explain the exclusion of non-peer-reviewed research, such as conference abstracts, preprints, etc., if any.
Response 2: Thank you for your thoughtful comment. We have clarified these points in Section 2.2 (Eligibility Criteria) of the revised manuscript, including the exclusion of case reports, scoping reviews, and non-peer-reviewed sources such as conference abstracts and preprints.
Comments 3: "New opportunities for cross-species comparisons" should explain why spatial transcriptomics is a good fit for cross-species comparisons, as stated in Line 56.
Response 3: We explained it in our revised manuscript.
Comments 4: PCC framework is defined, but not directly connected to eligibility or data analysis. Suggest clarifying how PCC shapes inclusion criteria (Line 117–130)
Response 4: We clarified in our revised manuscript.
Comments 5: The total exceeds the number of studies (e.g., 1,062 human + 619 animal = 1,681 > 1,398). Are some studies counted in both categories?
Response 5: Yes. In the figure 3 caption we clearly mentioned that Of the studies included, 1,062 employed human models and 619 employed animal models. Animal and human models or multiple animal models are often combined within one publication.
Comments 6: Thematic synthesis is mentioned, but examples are sparse. Add 1 or 2 sentences elaborating the most notable cross-cutting themes beyond oncology (e.g., emergence of multi-modal methods, spatial resolution trends, etc.).
Response 6: Thank you for this helpful suggestion. We agree that providing concrete examples enhances clarity. In our revised manuscript, we expanded Section 6.1 (Summary of Key Findings) to include notable cross-cutting themes, such as the emergence of multi-modal approaches and increasing emphasis on spatial resolution across various research domains.
Comments 7: The examined literature reports both human and murine (mouse) models (Mouse: Most used animal model, and Human: 1,062 research). Some species, such as dogs, rats, pigs, zebrafish, and non-human primates, are underrepresented. Although they are still in early stages, veterinary and cattle applications are essential to comparative medicine and One Health objectives.
Response 7: Thank you for your thoughtful comment. We have addressed this point in our revised manuscript by discussing the predominance of human and murine models and the underrepresentation of other species such as dogs, pigs, and non-human primates. This was added in Section 6.1 (Summary of Key Findings) and also acknowledged as a limitation in Section 6.4 (Research Gaps and Limitations of the Literature).
Comments 8: ST studies are divided into three categories: Mechanistic (the largest group), which includes immunological responses, cell signaling, development, and foundational biology; Therapeutic (drug response, treatment stratification, and personalized medicine); and Diagnostic (biomarker discovery, disease subtyping, and syndrome-linked profiling). Although mechanistic findings predominate, therapeutic and diagnostic applications, particularly in oncology, are growing.
Response 8: We fixed this in our revised manuscript.
Comments 9: "Spatial transcriptomics is poised to become a cornerstone..." (line 753): This is a strong claim. Consider softening or citing trends or predictions to support it.
Response 9: Thank you for the valuable suggestion. In our revised manuscript, we have softened the statement and added supporting examples and trends in the same section to justify the claim appropriately.
Comments 10: CIBERSORT is a cell type deconvolution tool, but Harmony is not, it's for batch correction. "CIBERSORT and Harmony are specialized tools for cell type deconvolution and corbatch effect correction, respectively [10,18,41]." Consider "CIBERSORT is used for cell type deconvolution, while Harmony is primarily used for batch effect correction."
"Leiden clustering, Principal Component Analysis (PCA), and Uniform Manifold Approximation and Projection (UMAP)..." These are methods/algorithms, not software tools per se. Consider "Analytical methods such as Leiden clustering, PCA, and UMAP..."
Response 11: We fixed these issues in our revised manuscript.
Minor revision
Line 30–35: “Spatial transcriptomics” is repeated redundantly within the same paragraph without adding clarity.
Response: We fixed this issue in our revised manuscript.
Line 53: "insights into veterinary medicine and animal health mucosal immunity" should be "insights into mucosal immunity related to veterinary medicine and animal health."
Response: We fixed this issue in our revised manuscript.
Line 54: “While limited in volume…” should be “Although limited in volume…” or “Despite being limited…”
Response: We fixed this issue in our revised manuscript.
Line 85: "spatial transcriptomics have been applied" should be "has been applied"
Response: We fixed this issue in our revised manuscript.
Line 116: "These objectives directly align..." Consider: “These objectives align with the rationale of the review by…”
Response: We fixed this issue in our revised manuscript.
Line 109–110: "integration with complementary technologies" Clarity would be improved by an example.
Response: We fixed this issue in our revised manuscript.
Line 131–134: "conducted in accordance with... Arksey and O’Malley, further refined by Levac et al..." Consider: “...based on the Arksey and O’Malley framework, refined by Levac et al., and reported using PRISMA-ScR guidelines...”
Response: We fixed this issue in our revised manuscript.
Line 193–196: Redundant listing of terms like "spatial transcriptomics," "spatial transcriptome profiling," "tissue-based transcriptomics" etc., could be more concise. Many of these are variations of the same term.
Response: Thank you for the suggestion. We agree that several of the search terms—such as “spatial transcriptomics,” “spatial transcriptome profiling,” and “tissue-based transcriptomics”—represent conceptual variants of the same methodology. However, we chose to retain all related terms in the Boolean search string to ensure comprehensive coverage and avoid missing relevant publications due to variations in keyword usage across studies.
The article uses both hyphenated ranges and comma-separated citations inconsistently. For instance, Line 38 uses the hyphenated range "[3-6]," Line 42 uses the comma-separated phrase "[5,7,8]," and Line 44 uses the en-dash with surrounding space "[8–10]." Carefully examine the text.
Response: We apologized for the mistake in our revised manuscript. We made all the corrections.
The claim that an adequate examination of spatial transcriptomics is lacking is frequently expressed without offering much new context. These could be consolidated.
Response: Thank you so much for your comments. In our revised manuscript we tried to avoid frequently repeating claims as much as we can.
Maynard et al. (2021) is cited four separate times (lines 665, 690, 704, and earlier), which may be intentional but appears redundant. Consider consolidating or cross-referencing the same citation for clarity.
Response: We clarify this issue in our revised manuscript.
Delorey et al. (2021) is referenced in multiple sections (lines 632 and 733). While valid, clarify whether these refer to the same study or different papers by the same authors.
Response: We clarify this issue in our revised manuscript.
In line 622, "[50]" is used to reference Liu et al. (2022), but line 657 also cites "[50]" for cardiac aging studies. If these are different studies, this indicates an error in citation indexing.
Response: Thank you very much for pointing this out. We carefully reviewed the citation and confirmed there was an error. We have now corrected this and removed the incorrect reference from line 733.
"Spatial transcriptomics’s growing role" (line 646): shold be: "The growing role of spatial transcriptomics."
Response: We fixed this issue in our revised manuscript.
"Psychiatric disorders... are increasingly recognized to involve..." (line 700): Consider rephrasing for conciseness: "...involve spatially defined disruptions..."
Response: We fixed this issue in our revised manuscript.
Line 737: Long sentence mixing multiple ideas. Could be split into two: one about canine osteosarcoma, another about its implications.
Response: We fixed this issue in our revised manuscript.
"These enhance veterinary diagnostics..." (line 738): Unclear what "these" refers to, should clarify.
Response: We fixed this issue in our revised manuscript.
Some sections begin with “disorders” (e.g., Metabolic Disorders, Neurodevelopmental Disorders), while others use “diseases” (Gastrointestinal Diseases, Cardiac Diseases). Suggest consistent usage: either “diseases” or “disorders” throughout, or match common clinical usage (e.g., “psychiatric disorders,” “infectious diseases”).
Response: We fixed this issue in our revised manuscript.
Seurat’s dominance is mentioned 3+ times in a short span. State this once clearly and refer back if needed (e.g., "As shown in Figure 10...").
Response: We fixed this issue in our revised manuscript.
“...deconvolute cell types.” consider “perform cell type deconvolution” or “identify cell types via deconvolution.”
Response: We fixed this issue in our revised manuscript.
“Analysis of spatial transcriptomics data involves custom adaptation of single-cell RNA-seq pipelines to preserve spatial context.” consider “Spatial transcriptomics analysis often adapts single-cell RNA-seq workflows to preserve spatial context, requiring integration of specialized tools tailored to both molecular data and tissue architecture.”
Response: We fixed this issue in our revised manuscript.
Round 2
Reviewer 1 Report
Comments and Suggestions for Authors
The author answered my question, and I find most of their response mostly acceptable with reservations. Regarding the newly added Section 6.1 in response, I have a question about that section. This entire section lacks references. Each discussed point requires supporting literature, as illustrated below:
- For exploratory mapping of tissues and construction of atlases, citations are needed for both mapping methodologies and atlas development studies.
- For applications in oncology, regenerative medicine, and infectious diseases, supporting articles must cover all three distinct fields.
- For biomarker validation, treatment stratification, and disease modeling, relevant literature should encompass each of these three research areas.
Reviewer 3 Report
Comments and Suggestions for Authors
The manuscript has been thoroughly revised according to the reviewers' suggestions, and the necessary changes have been made to significantly improve the clarity of the text. No further adjustments are required at this stage.